# Inference-time Alignment with Rewards in Anisotropic Besov Spaces: Superiority of Neural Networks over Linear Estimators

## Abstract

Inference-time alignment, the approach of adapting pre-trained models to rewards through reinforcement learning, has proven highly effective in enhancing the performance of language models. Despite its practical success, theoretical analysis remains underdeveloped, and in particular, only a limited number of studies address the practical setting where neural networks are employed as reward models. In this paper, we investigate the advantages of neural networks in inference-time alignment. Assuming that the true reward function lies in anisotropic Besov spaces, we derive upper bounds on the regret with respect to the number of oracle queries when using a neural network as a reward estimator. We further investigate the limitations of linear reward estimators, and show that neural networks are superior owing to their ability to adapt to the smoothness of functions. Finally, we demonstrate that, with an algorithm that iteratively and actively learns the reward model from the responses of the trained model, smaller regret can be achieved, as neural networks adapt to local structures.

## 1 Introduction

Inference-time compute (Brown et al., 2024; Snell et al., 2024; Wu et al., 2024b; OpenAI, 2024; Guo et al., 2025) has been attracting attention as a new paradigm for further enhancing the performance of pre-trained language models (LMs). By effectively leveraging the computational budget available at inference time, one can enhance the quality of model outputs without being restricted to pre-constructed datasets. A variety of techniques are included in this paradigm, e.g, long chains of thought (Wei et al., 2022; Li et al., 2024), self-evaluation and revision of own outputs (Zheng et al., 2023; Wu et al., 2024a), and exploration of improved responses (Yao et al., 2023; Zhang et al., 2024). Among these approaches, inference-time alignment, a framework to sample responses for LMs to maximize the reward via reinforcement learning, has been shown to offer a simple yet highly effective means of improving performance.

The methods for inference-time alignment has been widely studied from theoretical perspectives. For example, Yang et al. (2024); Beirami et al. (2025); Mroueh & Nitsure (2025) analyzed the performance of Best-of-$N$ alignment, which is the most basic method for inference-time alignment. Moreover, Huang et al. (2025a) pointed out the limitations of Best-of-$N$ alignment, proposed a new method based on $\chi^2$-divergence regularization. While these studies give insights on how each method is effective, their analysis mainly under the fixed reward model and do not incorporate the process of training the reward model. Foster et al. (2025) has analyze the training of reward models and show the advantage of multi-turn exploration method. However, their analysis focuses on the setting where the reward model is a linear estimator, which is far from practical settings where neural networks are used. This raises the following question:

> *What advantages do neural networks offer for inference-time alignment,*
> *and how can we unlock their full potential?*

More concretely, our question is how feature learning ability of neural networks can help the performance of inference-time alignment. Actually, to minimize the regret, we need to make our model's distribution concentrate around the optimal location. However, the optimal response can be located

on just a single point in a high dimension space, which makes deep learning more advantageous due to its feature learning ability. For that purpose, we consider an anisotropic Besov space (Nikol'skii, 1975; Vybiral, 2006; Triebel, 2011) as the model of the true reward, and see how neural network is effective to maximize the reward. Especially, we theoretically compare the performance of neural networks with that of *linear estimators* which is a class of estimators that cannot perform feature learning. Moreover, we consider a multi-step update of inference alignment in which we iteratively update our reward and policy models by observing reward oracles at each round. Then, we see how the regret will be improved by this multiple-update approach.

**Contributions.** Our contributions are summarized as follows:

1. **Regret bound for neural network reward estimator.** We derive an upper bound of the regret for inference-time alignment when the reward function lies in anisotropic Besov spaces. The anisotropic Besov space is a general function class that has different smoothness toward different directions. In addition to that, a function in the class has non-uniform smoothness over the input domain, which requires our estimator to perform feature learning to achieve the optimal estimation error rate (Suzuki & Nitanda, 2021). We utilize a regret bound by Huang et al. (2025a) that characterizes the regret bound by the squared loss error and the *coverage* which represents how large the pretrained generative model has mass around the maximum reward point (Jin et al., 2021; Xie et al., 2021; Zhu et al., 2023; Zhan et al., 2024; Li et al., 2023; Xiong et al., 2024).

2. **Superiority of neural networks against linear estimators.** We demonstrate that neural networks can adapt to local smoothness of the true reward function and generate responses with higher rewards compared to any linear estimator for approximating the reward model. We show sub-optimality of alignment methods based on a reward model estimated by a linear estimator by leveraging the fact that linear estimators cannot achieve optimal rate to estimate the reward function, while deep learning achieves faster rate. This highlights the advantage of feature learning ability by neural networks in reward maximization.

3. **Improved analysis of regret by multiple-step update.** We also analyze an algorithm that iteratively and actively learns the reward model from the responses of the trained model, and show that it achieves a smaller regret. Since our theoretical analysis requires boundedness of the coverage throughout the algorithm, we utilize a novel Gaussian perturbation technique. With the help of this method, we show that the regret is improved by multiple-step updates.

## 1.1 OTHER RELATED WORKS

**Capabilities of Neural Networks in Regression.** Theoretical analysis of neural networks and its superiority over other models has extensively studied in the context of regression problems. For example, Schmidt-Hieber (2020) and Suzuki (2018) showed that neural networks can achieve minimax optimal rates for estimating functions in Hölder spaces and Besov spaces, respectively. Suzuki & Nitanda (2021) extended the analysis to the case of anisotropic Besov spaces. They also showed the lower bounds on the estimation error for linear estimators, demonstrating the superiority of neural networks over linear estimators. Hayakawa & Suzuki (2020) also analyzed the upper bounds for neural networks and lower bounds for linear estimators, and showed that neural networks are superior to linear estimators for function classes with sparsity. Furthermore, Petersen & Voigtlaender (2018) and Imaizumi & Fukumizu (2019) analyzed the estimation error of neural networks for complicated functions with piecewise smoothness. Unlike these studies, our analysis focuses on the setting of inference-time alignment, which aims to find the response that maximizes the reward function, rather than minimizing the estimation error.

**Theoretical Analysis on Maximization of Black-box Functions.** Our study is highly related to the literature of black-box optimization. In particular, previous studies such as Minsker (2012), Minsker (2013), Grill et al. (2015), Wang et al. (2018) and Singh (2021) consider the setting where the objective function lies in RKHS, Hölder or Besov spaces, sometimes with additional assumptions on the structure of the function. While some of the techniques from these studies can be applied to our analysis, this paper differentiates itself in two aspects: (i) our analysis considers the setting of inference-time alignment, where the function to maximize is conditioned by a prompt; (ii) we assume some additional structure on the reward function, and demonstrate how the advantage of neural networks and multi-step training emerge.

## 1.2 NOTATIONS

Let $d_X, d_Y \in \mathbb{Z}_{>0}$ be the dimensions of prompts and responses, respectively, and $d = d_X + d_Y$. Let $\Omega_X = [0,1]^{d_X}, \Omega_y = [0,1]^{d_Y}, \Omega = [0,1]^d$. Let $\lambda$ be the Lebesgue measure on $\Omega$. For a function $f : \Omega \to \mathbb{R}$, let $\|f\|_p := \|f\|_{L^p(\Omega)} := \left( \int_\Omega |f|^p \mathrm{d}x \right)^{1/p}$ for $0 < p < \infty$, and $\|f\|_\infty := \|f\|_{L^\infty(\Omega)} := \sup_{x \in \Omega} |f(x)|$. For $\iota > 0$, a set $S$, a metric $\rho$, let $B(x, \iota; \rho)$ be the $\rho$-ball with center $x$ and radius $\iota$, and $\mathcal{M}(\iota; S, \rho)$ be the $\iota$-covering number of $S \subset \mathbb{R}^d$ with respect to $\rho$.

## 2 PROBLEM SETTINGS

### 2.1 INFERENCE-TIME ALIGNMENT

Inference-time alignment is a problem of generating a response $y \in \Omega_Y$ with high response for a given prompt $x \in \Omega_X$. More formally, let $P_X$ be a distribution on $\Omega_X$ and $\pi_{\mathrm{ref}}(y \mid x)$ be a base policy, which is typically a pre-trained language model. Let $r^\circ : \Omega \to [-R, R]$ $(R > 0)$ be a reward function that evaluates the quality of the response $y$ for the prompt $x$. We can only access the reward function via an oracle defined as

$$r^\dagger = r^\circ(x, y) + \xi, \quad \xi \sim \mathcal{N}(0, \sigma^2), \tag{1}$$

which returns a noisy observation of the reward for a given pair of prompt and response. Since the observation of reward is expensive (e.g., it requires human evaluation), we can only access a limited number of samples from the oracle.

The theoretical evaluation of inference-time alignment is based on *regret* defined as

$$J(\pi) := \mathbb{E}_{x \sim P_X} \left[ r^*(x) - \mathbb{E}_{y \sim \pi(\cdot \mid x)}[r^\circ(x, y)] \right],$$

where $r^*(x) := \max_{y \in \Omega_Y} r^\circ(x, y)$ is the maximum reward for the prompt $x$. The goal of inference-time alignment is to find a policy $\pi$ that minimizes the regret $J(\pi)$ for a fixed oracle size $n$.

As a technical assumption, we assume that it holds that $\pi_{\min} \le \pi_{\mathrm{ref}}(y \mid x) \le \pi_{\max}$ for all $x \in \Omega_X$ and $y \in \Omega_Y$, where $\pi_{\min}, \pi_{\max} > 0$ are universal constants.

### 2.2 DEFINITION OF ANISOTROPIC BESOV SPACE

In this paper, we assume that the reward function $r^\circ$ lies in an anisotropic Besov space. Roughly speaking, the anisotropic Besov space has a function class that has a different smoothness toward different directions. Feature learning ability plays essential role to estimate a function in this class because it is required to capture this anisotropic smoothness adaptively from data to achieve the optimal rate (Suzuki & Nitanda, 2021). We provide its formal definition here.

For a function $f : \mathbb{R}^d \to \mathbb{R}$, we define the *r-th difference of f in the direction* $h \in \mathbb{R}^d$ as

$$\Delta_h^r(f)(x) := \Delta_h^{r-1}(f)(x+h) - \Delta_h^{r-1}(f)(x), \quad \Delta_h^0(f)(x) := f(x),$$

for $x \in \Omega$ with $x + rh \in \Omega$, otherwise, let $\Delta_h^r(f)(x) = 0$.

**Definition 1** (Modulus of Smoothness). For a function $f \in L^p(\Omega)$ where $p \in (0, \infty]$, the $r$-th modulus of smoothness of $f$ is defined by $w_{r,p}(f, t) = \sup_{h \in \mathbb{R}^d : |h_i| \le t_i} \|\Delta_h^r(f)\|_p$, $t = (t_1, \ldots, t_d)$, $t_i > 0$.

In short, the modulus of smoothness is the $L^p$-norm of the $r$-th order finite derivative. With this modulus of smoothness, we define the anisotropic Besov space $B_{p,q}^{\boldsymbol{s}}(\Omega)$ for $\boldsymbol{s} = (s_1, \ldots, s_d)^\top \in \mathbb{R}_{>0}^d$ as follows.

**Definition 2** (Anisotropic Besov Space $B_{p,q}^{\boldsymbol{s}}(\Omega)$). For $0 < p, q \le \infty$, $\boldsymbol{s} = (s_1, \ldots, s_d)^\top \in \mathbb{R}_{>0}^d$, $r := \max_i \lfloor s_i \rfloor + 1$, let the seminorm $|\cdot|_{B_{p,q}^{\boldsymbol{s}}}$ be

$$|f|_{B_{p,q}^{\boldsymbol{s}}} := \begin{cases} \left( \sum_{k=0}^\infty \left[ 2^k w_{r,p}\left( f, (2^{-k/s_1}, \ldots, 2^{-k/s_d}) \right) \right]^q \right)^{1/q}, & (q < \infty), \\ \sup_{k \ge 0} 2^k w_{r,p}\left( f, (2^{-k/s_1}, \ldots, 2^{-k/s_d}) \right), & (q = \infty). \end{cases}$$

The anisotropic Besov space $B_{p,q}^{\boldsymbol{s}}(\Omega)$ is defined as $B_{p,q}^{\boldsymbol{s}}(\Omega) := \left\{ f \in L^p(\Omega) \mid \|f\|_{B_{p,q}^{\boldsymbol{s}}} < \infty \right\}$. where the norm $\|\cdot\|_{B_{p,q}^{\boldsymbol{s}}(\Omega)}$ is defined by $\|f\|_{B_{p,q}^{\boldsymbol{s}}} := \|f\|_p + |f|_{B_{p,q}^{\boldsymbol{s}}}$.

Intuitively, the parameter $\boldsymbol{s}$ represents the smoothness of each coordinate of the function. If $s_i$ is large, then the function is smooth in the $i$-th coordinate. When $s_1 = \cdots = s_d = s$, the anisotropic Besov space $B_{p,q}^{\boldsymbol{s}}(\Omega)$ matches with the Besov space $B_{p,q}^{s}(\Omega)$ (DeVore & Popov, 1988; DeVore et al., 1993). Moreover, $p = q = \infty$, then $B_{p,q}^{s}(\Omega)$ coincides with the Hölder space $C^s(\Omega)$ (Triebel, 2011). The parameter $p$ represents *uniformity* of the smoothness over the input space $\Omega$. We see that, when $p$ is small, the smoothness of functions in the class is guaranteed only in a average sense over the domain $\Omega$, hence the function can have a bumpy shape around some input point $x$. The feature learning ability plays a crucial role to detect such a bumpy point to achieve the optimal rate (Suzuki, 2018).

Throughout this paper, for the smoothness parameter $\boldsymbol{s} \in \mathbb{R}_{>0}^d$, let $\widetilde{s} := \left( \sum_{j=1}^d 1/s_j \right)^{-1}$, $\overline{s} := \max_{j=1,\ldots,d} s_j$, and $\underline{s} := \min_{j=1,\ldots,d} s_j$. We can regard $\widetilde{s}$ as the "total smoothness" that summarizes the smoothness toward all directions. Moreover, let $\rho_{\boldsymbol{s},p}$ ($p \in [1,\infty)$) be the metric on $\mathbb{R}^d$ defined by $\rho_{\boldsymbol{s},p}(x,y) := \left( \sum_{j=1}^d |x_j - y_j|^{ps_j/\underline{s}} \right)^{\underline{s}/(p\overline{s})}$ for $x, y \in \mathbb{R}^d$. We also define $\rho_{\boldsymbol{s},\infty}(x,y) := \left( \max_{j=1,\ldots,d} |x_j - y_j|^{s_j/\underline{s}} \right)^{\underline{s}/\overline{s}} = \max_{j=1,\ldots,d} |x_j - y_j|^{s_j/\overline{s}}$ for $x, y \in \mathbb{R}^{d1}$.

# 3 SUPERIORITY OF NEURAL NETWORKS OVER LINEAR ESTIMATORS

In this section, we consider a single-step update method for inference-time alignment. For the alignment, we use the `InferenceTimePessimism`(Huang et al., 2025a) in which we generate responses following an updated distribution which is constructed so that it has higher probability for responses with higher estimated rewards. Here, we utilize the neural network to estimate the reward function, and we freeze the reward function once it is estimated. In that sense, we say it is *single-step* update. To find higher reward responses, we need to estimate the reward function as accurate as possible. Indeed, we show that neural networks can achieve a smaller regret compared to linear estimators because the neural network achieves higher accuracy in estimating the reward, in which the non-uniformity of the smoothness of the anisotropic Besov space plays the essential role. Here, a linear estimator is a class of estimators that cannot perform nonlinear feature learning depending on the output $(y_i)_{i=1}^n$

For the analysis, we put the following assumption.

**Assumption 3.** *For a reward function $r^\circ \in B_{p,q}^{\boldsymbol{s}}(\Omega)$ ($p,q \in [1,\infty]$, $\boldsymbol{s} \in \mathbb{R}_{>0}^d$, $\widetilde{s} > 1/p$), we define $S_\epsilon(x) := \{ y \mid r^*(x) - r^\circ(x,y) \le \epsilon \}$ and $\mathcal{S}_\epsilon := \{ (x,y) \mid r^*(x) - r^\circ(x,y) \ge \epsilon \}$. Let $\gamma \in [0, \frac{1}{\overline{s}-1/p})$ and $\epsilon_0 > 0$ be constants. Then, we assume that it holds $\lambda(S_\epsilon(x)) \gtrsim \epsilon^\gamma$ for all $\epsilon \in (0, \epsilon_0]$ and $x \in \Omega_X$.*

Assumption 3 assumes that the super-level set has a sufficiently large volume. Technically, this assumption guarantees that there exists a comparator policy with small coverage (See Lemma 19 for details). We will prove that neural networks can capture the large super-level set, thereby achieving a small regret (Theorem 4), while linear models cannot (Theorem 5).

## 3.1 UPPER BOUND OF REGRET FOR NEURAL NETWORK REWARD ESTIMATORS

We first present the upper bound of the regret that can be achieved by neural networks. Due to the feature learning ability of neural networks, we obtain better estimation of the reward so that we can achieve better regret.

**Class of Neural Networks.** To obtain the regret bound for neural network estimators of the reward, we formally define the class of neural networks used in this paper. Let $\eta := \max\{0, \cdot\}$ be the ReLU activation function. Then, a neural network with depth $L$ and width $W$ is defined as

$$f(x) = (A_L \eta(\cdot) + b_L) \circ \cdots \circ (A_2 \eta(\cdot) + b_2) \circ (A_1 x + b_1),$$

where $A_i \in \mathbb{R}^{d_{i+1} \times d_i}, b_i \in \mathbb{R}^{d_{i+1}}$ for $i \in [L]$ with $d_1 = d, d_{L+1} = 1$, and $\max_i d_i \le W$. Then, we define the class $\Phi(L, W, S, B)$ of neural networks with depth $L$, width $W$, sparsity $S$ and norm

---

[1]The map $\rho_{\boldsymbol{s},p}$ ($p \in [1,\infty]$) is indeed a metric. See Lemma 22 for the proof.

---

**Algorithm 1** Inference-Time Pessimism (`InferenceTimePessimism`$(x, \pi, \widehat{r}, N, \mu)$)

---

**Input:** Prompt $x$, policy $\pi$, reward model $\widehat{r}$, sample size $N$, regularization $\mu$.
1: Draw i.i.d. samples $y_1, \dots, y_N \sim \pi(\cdot \mid x)$.
2: Compute normalization constant $\widehat{\theta}(x)$ such that $\frac{1}{N} \sum_{i=1}^{N} [\widehat{r}(x, y_i) - \widehat{\theta}(x)]_+ = \mu$.
3: Set $M := \mu^{-1}(R - \widehat{\theta}(x))$ and and $w(y \mid x) := \mu^{-1}[\widehat{r}(x, y) - \widehat{\theta}(x)]_+$.
4: Sample $y$ as $y \sim \mathtt{RejectionSampling}_{N,M}(w; \pi_{\mathrm{ref}}, x)$.
5: **return:** response $y$.

---

bound $B$ as

$$\Phi'(L, W, S, B) := \left\{ f \;\middle|\; \max_i \{\|A_i\|_\infty, \|b_i\|_\infty\} \le B, \; \sum_{i=1}^{L} (\|A_i\|_0 + \|b_i\|_0) \le S \right\},$$

where $\| \cdot \|_\infty$ is the maximum absolute value of the entries ($\ell^\infty$-norm as a vector) and and $\| \cdot \|_0$ is the number of non-zero elements ($\ell^0$-norm as a vector). The $\ell^0$-norm constraint imposes sparsity of the model that controls the complexity of the model appropriately. Due to the technical convenience to analyze the estimation error, we consider the class of clipped neural networks defined as $\Phi(L, W, S, B) := \{\min\{\max\{f, -R\}, R\} \mid f \in \Phi'(L, W, S, B)\}$. Since the clipping function can be realized by ReLU units, this setting is not far from practical scenarios.

**Algorithm and Theoretical Guarantee.** Here, we present how to generate the responses with higher reward through the reward estimation. First, we generate $n$ input-prompts $x_1, \dots, x_n$ i.i.d. from $P_X$, and for each $i \in [n]$, we generate the responses $y_i \sim \pi_{\mathrm{ref}}(\cdot \mid x_i)$ from our pretrained reference model. Then, we observe noisy reward oracles as $r_i^\dagger := r^\circ(x_i, y_i) + \xi_i$ as in (1), where $\xi_i \sim \mathcal{N}(0, \sigma^2)$ is the observation noise. Then, we fir the neural network model to the observed reward by empirical risk minimization:

$$\widehat{r} := \underset{r \in \Phi(L, W, S, B)}{\arg\min} \sum_{i=1}^{n} (r_i^\dagger - r(x_i, y_i))^2,$$

where $L, W, S, B$ will be set appropriately depending on the smoothness of the true regret and the data size. Here, we denote by $D^n = \{(x_i, y_i)\}_{i=1}^n$. Using the estimated reward $\widehat{r}$, we update the generative model in accordance to the reward. For $\mu > 0$ (which can be dependent on $x$), we define $\pi_\mu^\chi$ by

$$\pi_\mu^\chi(\cdot \mid x) := \underset{\pi:\text{density on } \Omega_Y}{\arg\max} \; \mathbb{E}_{y \sim \pi}[\widehat{r}(x, y)] - \mu \cdot \chi^2(\pi \parallel \pi_{\mathrm{ref}}(\cdot \mid x)),$$

where $\chi^2(\cdot \| \cdot)$ is the $\chi^2$-square divergence defined as $\chi^2(\mu \| \nu) := \frac{1}{2} \mathbb{E}_\nu \left[ \left( \frac{d\mu}{d\nu} - 1 \right)^2 \right]^2$. Then, we can write $\pi_\mu^\chi$ in a closed form as

$$\pi_\mu^\chi(y \mid x) = \pi_{\mathrm{ref}}(y \mid x) \left[ \mu^{-1}(\widehat{r}(x, y) - \theta_\mu) \right]_+,$$

where $\theta_\mu$ is the normalizing constant such that $\int \pi_\mu^\chi(y \mid x)\,dy = 1$. `InferenceTimePessimism` (Algorithm 1) is a practical algorithm to get samples from $\pi_{\mu,N}^{\mathtt{Pes}}$ that approximates $\pi_\mu^\chi$, where $N \in \mathbb{Z}_{>0}$ is a function that determines the number of samples to be drawn from $\pi_{\mathrm{ref}}(\cdot \mid x)$.

Then, the response $\widehat{y}_{\mathrm{NN}}(x)$ for a prompt $x$ is generated by $\widehat{y}_{\mathrm{NN}}(x) \sim \pi_{\mu,N}^{\mathtt{Pes}}(\cdot \mid x)$.

**Theorem 4.** *Suppose that we set the parameters of the network as $L = O(\log N), W = O(N \log N), S = O(N \log^2 N), \log B = O(\log N)$ for $N = n^{\frac{1}{2\widetilde{s}+1}}$ sufficiently large. Then, under Assumption 3, the estimator $\widehat{y}_{\mathrm{NN}}$ satisfies*

$$\mathbb{E}_{D^n} \left[ \mathbb{E}_{x \sim P_X} \left[ \mathbb{E}_{\widehat{y}_{\mathrm{NN}}(x) \sim \pi_{\mu,N}^{\mathtt{Pes}}} [r^*(x) - r^\circ(x, \widehat{y}_{\mathrm{NN}}(x))] \right] \right] \lesssim n^{-\frac{2\widetilde{s}}{2\widetilde{s}+1} \cdot \frac{2}{2+\gamma}}.$$

---

[2]Huang et al. (2025b) showed that $\chi^2$-divergence provides more robust estimate to over-optimization and then better regret than the usual KL-divergence regularization. Hence, we also employ $\chi^2$-divergence in this paper.

It is known that $n^{-\frac{2\bar{s}}{2\bar{s}+1}}$ is the minimax optimal rate (Suzuki & Nitanda, 2021) in terms of $L^2$-norm to estimate a function in the anisotropic Besov space. The regret bound is slower than this rate up to $O(n^{\frac{2}{2+\gamma}})$. This difference is a cost to convey the $L^2$-norm error to the error to find the maximum of the true reward. However, due to the volume condition of the upper level set (Assumption 3), the $L^2$-norm estimate can be converted to $L^\infty$-norm type bound locally around the global optimal point.

The key of the proof of this theorem is the regret bound given by (Huang et al., 2025a) that characterizes the balance between the reward estimation error and the *coverage* of the reference measure. Let $\epsilon_{\mathtt{RM}}^2(x) := \mathbb{E}_{y\sim\pi_{\mathrm{ref}}(\cdot|x)}[(\widehat{r}(x,y) - r^\circ(x,y))^2]$ be the $L^2$-estimation error of our reward estimator $\widehat{r}$. For two policies, we define the coverage between them as

$$\mathcal{C}(x;\pi_1,\pi_2) := \mathbb{E}_{y\sim\pi_1(\cdot|x)}\left[\frac{\pi_1(y|x)}{\pi_2(y|x)}\right].$$

Then, for any comparator policy $\pi^*$, it holds that `InferenceTimePessimism` satisfies

$$\mathbb{E}_{y\sim\pi^*}[r^\circ(x,y)] - \mathbb{E}_{y\sim\pi_{\mu,N}^{\mathtt{Pes}}(\cdot|x)}[r^\circ(x,y)]$$

$$\lesssim \mu\cdot\mathcal{C}(x;\pi^*,\pi_{\mathrm{ref}}) + \mu^{-1}\cdot\epsilon_{\mathtt{RM}}^2(x) + \mu^{-1}\cdot\epsilon_{\mathtt{RM}}(x)\exp\left(-\frac{\mu N}{C_1(R+\mu)}\right), \qquad (2)$$

for any $\mu > 0$ (Huang et al., 2025a). From this relationship, we see trade-off between the estimation error $\epsilon_{\mathtt{RM}}^2$ and the coverage. To obtain a better regret, the reference model $\pi_{\mathrm{ref}}$ should "cover" a region around the maximum reward point and the reward function should be estimated accurately. As we will see in the next section, deep neural network attains better estimate than the linear model that gives advantage to neural network for achieving smaller regret.

## 3.2 LIMITATION OF LINEAR ESTIMATORS

Next, we compare the bound obtained in the last section with that of the linear estimators. The *linear estimator* is a class of estimators that can be written as

$$\widehat{f}(x) = \sum_{i=1}^{n} y_i\varphi_i(x;X^n),$$

where $X^n := (x_1,\ldots,x_n)$, and $\varphi_i(\cdot;X^n)$ are measurable functions that depend on $x$ and $X^n$ but not on $y_1,\ldots,y_n$. This estimator includes wide range of estimators such as $k$-NN regression, kernel ridge regression with a fixed kernel function, and sieve estimators. The linear estimator cannot calculate nonlinear effect from the output and thus cannot conduct nonlinear feature learning depending on the output $y$ (while it is allowed to conduct feature learning merely depending on input as performed in PCA). This difference induces the following sub-optimal rate.

**Theorem 5** (Limitation of Linear Estimators). *For any $\delta > 0$, there exists a set $\mathcal{F}_\delta$ of reward functions that is a subset of reward functions satisfying Assumption 3 such that the following holds.*

*(i)* $\inf_{\widehat{f}:linear}\sup_{f^\circ\in\mathcal{F}_\delta}\mathbb{E}_{D_n}\left[\left\|\widehat{f} - f^\circ\right\|_{L^2(P_X)}^2\right] \gtrsim n^{-\frac{2\bar{s}-v}{2\bar{s}+1-v}}$, *where* $v := 2(1/p - 1/2)_+$;

*(ii) There exists a function* $g \in L^2(\Omega)$ *such that* $\|g - f\|_{L^2(\Omega)} = \delta$ *for all* $f \in \mathcal{F}_\delta$;

*(iii)* $\max_{y\in\Omega_Y} r^\circ(x,y) - g(x,y) = \delta^{\frac{1}{2+\gamma}}$.

The item (i) implies that this lower bound of estimation rate matches the lower bound for $B_{p,q}^{\boldsymbol{s}}$ shown in Suzuki & Nitanda (2021). This indicates that the assumption of the volume of super-level set does not help improve the estimation rate of linear estimators. Moreover, item (ii) and (iii) imply that it is possible that the estimator with $L^2$ error less than $\delta$ cannot distinguish the functions in $\mathcal{F}_\delta$, and the regret can be as worse as $\delta^{1/(2+\gamma)}$. In particular, if $\delta$ is the estimation error of linear estimators, i.e., $\delta \simeq n^{-\frac{2\bar{s}-v}{2\bar{s}+1-v}}$, the regret can be $n^{-\frac{2\bar{s}-v}{2\bar{s}+1-v}\cdot\frac{1}{2+\gamma}}$ in the worst case, which is slower than the rate of neural networks. This particularly due to the fact that the linear estimators cannot perform future learning. The sub-optimality appears especially when $p$ is small, that is, the target function has a bumpy shape around some point $x$. The linear estimator is not as good as neural networks to adaptively find such a bumpy location due to lack of feature learning ability, which leads to the sub-optimal rate as shown above.

---

**Algorithm 2** Multi-step Training for the Reward Model (`MultiStepAlignment`$(\pi_{\mathrm{ref}}, n)$)

**Input:** Base policy $\pi_{\mathrm{ref}}$, size of oracle queries $n$.

1: Set $\pi^{(0)} := \pi_{\mathrm{ref}}$, $T := \lceil \log n \rceil$, $n_0 := \lfloor n/T \rfloor$.
2: **for** $\tau = 1, \ldots, T$ **do**
3:      Set the hyperparameters $N^{(\tau)}, \mu^{(\tau)}, \sigma^{(\tau)}$.
4:      Draw $n_0$ samples $\{x_t\}_{t=(\tau-1)n_0+1}^{\tau n_0} \sim P_X$.
5:      For each $t = (\tau-1)n_0+1, \ldots, \tau n_0$, draw $y_t \sim \pi^{(\tau-1)}(\cdot \mid x_t)$.
6:      Observe the reward $\{r_t^\dagger\}_{t=(\tau-1)n_0+1}^{\tau n_0}$ for $\{(x_t, y_t)\}_{t=(\tau-1)n_0+1}^{\tau n_0}$ using the oracle (1).
7:      Get the set of indices $\mathcal{T}_\tau := \{t \mid (\tau-1)n_0 + 1 \le t \le \tau n_0, y_t \in \Omega_Y\}$.
8:      Train the reward model $\widehat{r}^{(\tau)} := \arg\min_{r \in \Phi(L,W,S,B)} \sum_{t \in \mathcal{T}_\tau} (r(x_t, y_t) - r_t^\dagger)^2$.
9:      Set $\pi^{(\tau)} \leftarrow \pi^{\mathtt{M\text{-}Pes}}[\pi^{(\tau-1)}, \widehat{r}^{(\tau)}, N^{(\tau)}, \mu^{(\tau)}, \sigma^{(\tau)}]$    # $\pi_{\mu,N}^{\mathtt{Pes}} + \mathcal{N}(0, (\sigma^{(\tau)})^2 I_{d_Y})$
10: **end for**
11: **return:** policy $\pi^{(T)}$.

---

## 4   Inference-time Alignment with Neural Networks

In this section, we propose a multi-step algorithm for inference-time alignment while we considered a single-step method in the previous section. By extending the algorithm to multi-step, we can make use of a stronger assumption on the regret so that we obtain a better regret bound. We also provide a theoretical guarantee of the regret bound for the proposed algorithm with respect to the size of oracle queries $n$.

### 4.1   Algorithm: Multi-step Training for the Reward Model

The concrete procedure of our proposed algorithm is described in Algorithm 2. We also provide an illustrative explanation in the right part of Figure 1. Basically, it repeats the alignment method in the previous section multiple times. However, as we have seen in (2), our policy should have a small coverage $\mathcal{C}(x; \pi^*, \pi_{\mathrm{ref}})$. When we update our policy multiple-times, it is expected that our policy will "concentrates" around the maximum reward point. To achieve this, we iteratively update the reward function and sampling.

In each step $\tau$, we generate $n_0$ query points $\{(x_t, y_t)\}_{t=(\tau-1)n_0+1}^{\tau n_0}$ from our current generative model $\pi^{(\tau-1)}$:

$$\widehat{\pi}^{(\tau)}(y|x) = \pi^{(\tau-1)}(y|x)\Big[(\widehat{r}^{(\tau)}(x,y) - \theta_\mu^{(\tau)})/\mu^{(\tau)}\Big]_+,$$

for the reward estimate $\widehat{r}^{(\tau)}$ at $\tau$-th round, where $\mu^{(\tau)}$ is set appropriately. However, we only have $L^2$-norm guarantee of our reward estimate $\widehat{r}^{(\tau)}$, which is not sufficient to bound the coverage $\mathcal{C}(x; \pi^*, \widehat{\pi}^{(\tau)})$. For that purpose, we mollify the density $\widehat{\pi}^{(\tau)}$ of our policy by adding the Gaussian noise $\mathcal{N}(0, (\sigma^{(\tau)})^2 I_{d_Y})$ to each generated point so that the distribution of our generated points can cover the maximum reward point with non-vanishing probability. (The resulting distribution is referred to as $\pi^{\mathtt{M\text{-}Pes}}$ in Algorithm 2.) This guarantees a bound on the coverage and then we obtain a proper convergence of regret as shown below. The noise scale is set to $\sigma^{(\tau)} \simeq n_0^{-\frac{1}{2+\gamma}\frac{2\widetilde{s}}{2\widetilde{s}+1}\frac{1-u^\tau}{1-u}}$, where $u > 0$ is a constant defined in Theorem 7. This choice trades off exploitation of the current reward model against sufficient coverage: if $\sigma^{(\tau)}$ is too small, sampling becomes overly concentrated around potentially biased maximizers and misses regions of high true reward; if it is too large, samples become nearly independent of the learned model and fail to leverage the information gathered so far.

### 4.2   Theoretical Guarantee

Now, we give the regret bound for the multi-step algorithm (Algorithm 2). Since we need a bound on the coverage during the update, we put the following assumption.

**Assumption 6.** *For a reward function* $r^\circ \in B_{p,q}^{\boldsymbol{s}}(\Omega)$ *($p, q \in [1, \infty], \widetilde{s} > 1/p$), we define* $\mathcal{S}_\epsilon := \{(x, y) \mid r^*(x) - r^\circ(x, y) \ge \epsilon\}$. *We use the following assumptions.*

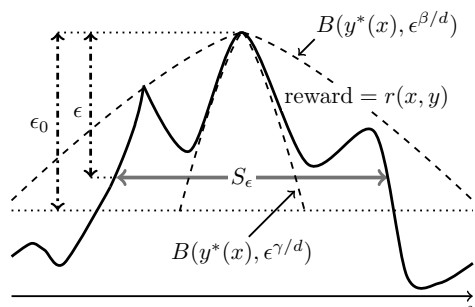 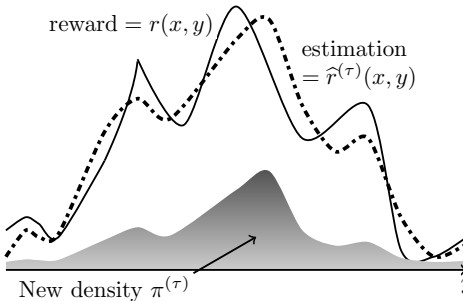

Figure 1: Conceptual illustrations of our assumption and algorithm. (Left) In Assumption 6, we impose assumptions on the local landscape of the reward around the maximum point. Specifically, we assume that, for all $\epsilon \in (0, \epsilon_0]$, the super-level set $S_\epsilon(x)$ satisfies $B(y^*(x), \epsilon^{\gamma/d}) \subseteq S_\epsilon(x) \subseteq B(y^*(x), \epsilon^{\beta/d})$. We remark that, when $p$ is small, our assumption allows locally bumpy shapes of the reward function (as in the figure), since the anisotropic Besov space $B_{p,q}^s$ includes such functions. (Right) Our multi-step algorithm (Algorithm 2) mainly consists of two procedures: first, we estimate the reward model $r$ by $\widehat{r}^{(\tau)}$ using samples from $\pi^{(\tau-1)}$. Then, we obtain the updated density $\pi^{(\tau)}$, which prioritizes responses $y$ with high estimated rewards. Hence, in the next step $\tau + 1$, the estimation of the reward is more accurate around the maximum point, which results in a higher expected reward of the responses generated from $\pi^{(\tau+1)}$.

**(A1)** *There exists constants* $\epsilon_0 > 0$ *and* $\beta, \gamma$ *with* $0 \le \beta \le \gamma \le \widetilde{s} - 1/p$ *such that* $B(y^*(x), \epsilon^{\gamma/d}) \subseteq S_\epsilon(x) \subseteq B(y^*(x), \epsilon^{\beta/d})$ *for all* $\epsilon \in (0, \epsilon_0]$ *and* $x \in \Omega_X$, *where* $y^*(x) := \arg\max_{y \in \Omega_Y} r^\circ(x, y)$.

**(A2)** *There exist constants* $c_0, C_0 > 0$ *such that* $\mathcal{M}(\delta; \mathcal{S}_\epsilon, \rho_{\boldsymbol{s}, 2}) \le C_0 \left(1 + \frac{\lambda(\mathcal{S}(\epsilon))}{V_d(\delta)}\right)$ *for all* $\epsilon, \delta \in (0, c_0]$, *where* $V_d(\delta) \simeq \delta^{\overline{s}/\widetilde{s}}$ *is the volume of* $\rho_{\boldsymbol{s}, 2}$*-ball with radius* $\delta$.

**Remark.** **(A1)** requires that the super-level set be concentrated around the maximizer, with both lower and upper bounds imposed on the distance from the maximizer. The lower bound on this distance is necessary for the algorithm to capture the rough location of the super-level set, while the upper bound is required to narrow down the position of $y^*(x)$ within the super-level set. A simple example is the case where $r^\circ(x, \cdot)$ is locally strongly convex around $y^*(x)$. In this case, the assumption holds with $\beta = \gamma = d/2$. **(A2)** is an assumption brought from Wang et al. (2018), which is a literature on the oracle complexity for optimization of Hölder smooth functions. This imposes a regularity condition of the set $\mathcal{S}_\epsilon$. This assumption is satisfied, for example, when $\mathcal{S}_\epsilon$ is a finite union of $\rho_{\boldsymbol{s}, 2}$-balls. See also the left part of Figure 1.

Then, we have the following regret bound for our algorithm.

**Theorem 7.** *Let* $r^\circ \in B_{p,q}^{\boldsymbol{s}}(\Omega)$ *be the reward function, and assume that* $\boldsymbol{s} \in \mathbb{R}_{>0}^d$, $p, q \in [1, \infty]$, $\widetilde{s} \ge 1/p$. *Moreover, we assume that* $r^* \in B_{p,q}^{\boldsymbol{s}}(\Omega_X)$, $\gamma > 2$ *and* $\beta \in [0, 1/2(\widetilde{s} - 1/p))$. *Additionally, assume that it holds, for any $x$ and step $\tau$, it holds* $\mathbb{E}_y\left[(\widehat{r}^{(\tau)}(x, y) - r^\circ(x, y))^2\right] \le C\mathbb{E}_{x \sim P_X}\mathbb{E}_y\left[(\widehat{r}^{(\tau)}(x, y) - r^\circ(x, y))^2\right]$ *for some constant* $C > 0$. *Then, under (A1) and (A2), the output $\pi^{(T)}$ of Algorithm 2 satisfies*

$$\mathbb{E}_{x \sim P_X}[r^*(x) - \mathbb{E}_{y \sim \pi^{(T)}}[r^\circ(x, y)]] \lesssim \left(\frac{\log n}{n}\right)^{\frac{1}{2+\gamma}\frac{2\widetilde{s}}{2\widetilde{s}+1}\frac{1}{1-u}} \text{poly}\log(n),$$

*where* $u := \frac{\alpha\beta}{2+\gamma}\frac{2\beta}{d}\left(\frac{1}{2} + \frac{2\varsigma}{2\widetilde{s}+1}\right)$, $\varsigma \in (0, \widetilde{s} - 1/p)$ *and* $\alpha := \min(1, \widetilde{s} - 1/p)$ *are constants.*

This theorem implies that by learning the reward estimator using multi-step as shown in Algorithm 2, the regret with respect to the oracle size $n$ is improved by a factor of $\frac{1}{1-u}(> 1)$. This factor depends

on $\beta$, which represents the smallness of the super-level set of the reward $r^\circ(x, \cdot)$. As $\beta$ increases, the regret rate also improves. From this observation, it follows that through multi-step training, the neural-network-based reward estimator is able to capture the super-level set effectively.

### 4.3 PROOF SKETCH

The key factor of proof of Theorem 7 is to show that neural networks can adapt to the small super-level set of $r^\circ$. The following lemma indeed demonstrates this fact.

**Lemma 8** (Estimation Error under Large Expected Reward). *Assume that the reward function $r^\circ \in B_{p,q}^s(\Omega)$ and the distribution $P_X$ satisfy the same conditions as in Theorem 7. Moreover, suppose that $\pi$ is a policy satisfying $\mathbb{E}_{x \sim P_X}[r^*(x) - \mathbb{E}_{y \sim \pi(\cdot | x)}[r^\circ(x, y)]] \leq \mathcal{R}$. Let $D_n = \{(x_i, y_i, r_i^\dagger)\}_{i=1}^n$ be a dataset where $x_i \sim P_X$, $y_i \sim \pi(\cdot \mid x_i)$, and $r_i^\dagger = r^\circ(x_i, y_i) + \xi_i$ with $\xi_i \sim \mathcal{N}(0, \sigma^2)$. Then, under (A1)–(A4), the estimator $\widehat{r}$ of $r^\circ$ defined as $\widehat{r} := \arg\min_{r \in \Phi(L,W,S,B)} \sum_{i=1}^n (r(x_i, y_i) - r_i^\dagger)^2$, satisfies*

$$\mathbb{E}_{D_n}\left[\|\widehat{r} - r^\circ\|_{L^2(P_X \otimes \pi)}^2\right] \lesssim \mathcal{R}^{\frac{2\beta\varsigma}{2\tilde{s}+1}} \cdot n^{-\frac{2\tilde{s}}{2\tilde{s}+1}} \log^4(n),$$

*where $\mathbb{E}_{D_n}$ is the expectation with respect to the dataset $D_n$.*

When $\beta = 0$, the above lemma matches the existing results on the convergence rate of regression by neural networks for anisotropic Besov space (Suzuki & Nitanda, 2021) (up to $\log$-factors). We can see that if $\beta$ becomes larger and the super-level set of $r^\circ$ becomes smaller, the estimation error rate improves. By using this lemma, we can prove that during multi-step training, at each step, both the improvement of the regret $\mathcal{R}$ and the improvement of the reward estimation rate are repeated. Ultimately, after $\log n$ steps, the rate in the Theorem 7 is achieved.

## 5 CONCLUSION

This paper gives a convergence analysis of neural networks for test-time alignment problem; reward maximization. We consider a setting where the true reward is in an anisotropic Besov space where a function in the function class has non-uniform smoothness over the input space and toward different directions. Due to the feature learning ability of neural networks, it was shown that `InferenceTimePessimism` with neural networks can outperform a linear estimator based approach in terms of regret when the uniformity of the smoothness $p$ is small. In addition to that, we proposed a multiple-step update method for test-time alignment, and analyzed the regret bound for this method. Under an assumption that the super-level set of the reward is concentrated around the maximizer, we showed that the multiple-step method can improve the regret by refining the estimate of the location of the reward maximizer.

**Limitation and Future Work.** Although we showed improvement of regret by the multiple-step update in Theorem 7, we imposed a rather strong condition $\mathbb{E}_y\left[(\widehat{r}^{(\tau)}(x,y) - r^\circ(x,y))^2\right] \leq C\mathbb{E}_{x \sim P_X}\mathbb{E}_y\left[(\widehat{r}^{(\tau)}(x,y) - r^\circ(x,y))^2\right]$. This condition was used to covey a expected squared loss to a uniform bound to uniformly upper-bound the coverage. An interesting future work is to relax this condition or propose a new method to overcome this difficulty.

## LLM USAGE STATEMENT

LLM were used solely for editing and refining the writing, including correcting grammar and improving sentence structure. They were not used to generate any original content or ideas, nor deriving the proofs.

## ETHICS AND REPRODUCIBILITY STATEMENTS

This work is purely theoretical and has no ethical concerns. For reproducibility, we stated all assumptions in the main text and provided all proofs in the appendix.

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

# —— Appendix ——

## A  ADDITIONAL NOTATIONS

For $s \in \mathbb{R}^d_{>0}$, let $|s| := \sum_{j=1}^d |s_j|^2$ and $s' = [s'_j]_{j=1}^d := [\underline{s}/s_j]_{j=1}^d$. For $s \in \mathbb{R}^d_{>0}$ and $k \in \mathbb{Z}$, let $\|k\|_{s'} := \sum_{j=1}^d \lfloor ks'_j \rfloor$.

Let $\mathcal{K} : \mathbb{R} \to \mathbb{R}$ and $\mathcal{K}_m : \mathbb{R} \to \mathbb{R}$ be functions defined as

$$\mathcal{K}(x) = \begin{cases} 1 & \text{if } x \in [0,1], \\ 0 & \text{otherwise}, \end{cases}$$

$$\mathcal{K}_m(x) = (\underbrace{\mathcal{K} * \cdots * \mathcal{K}}_{m+1 \text{ times}})(x),$$

where $f * g(x) := \int f(x-t)g(t)\mathrm{d}t$ is the convolution of functions $f$ and $g$. The function $\mathcal{K}_m$ is called the *cardinal B-spline of order* $m$. Then, for $k \in \mathbb{N}_{>0}$ and $j = (j_1, \ldots, j_d) \in \mathbb{Z}^d$, let $M^d_{k,j} : \mathbb{R}^d \to \mathbb{R}$ be the function defined as

$$M^d_{k,j}(x) := \prod_{i=1}^d \mathcal{K}_m(2^{\lfloor ks'_i \rfloor} x_i - j_i),$$

Intuitively, the integer $k$ controls the spacial resolution, and $j$ controls the location of the function.

We also remark that the support of $M^d_{k,j}$ is the hyper-rectangle written by

$$\mathrm{supp}(M^d_{k,j}) = \prod_{i=1}^d \left[ 2^{-\lfloor ks'_i \rfloor} j_i, 2^{-\lfloor ks'_i \rfloor}(j_i + m + 1) \right].$$

Moreover, let $J(k)$ be the set of $j \in \mathbb{Z}^d$ such that $\mathrm{supp}(M^d_{k,j}) \cap \Omega \neq \emptyset$, i.e.,

$$J(k) := J_1(k) \times \cdots \times J_d(k),$$

where

$$J_i(k) := \{-m, -m+1, \ldots, 2^{\lfloor ks'_i \rfloor} - 1, 2^{\lfloor ks'_i \rfloor}\}.$$

## B  PROOF OF THEOREM 5

We first introduce the following proposition, which is convenient to analyze the limitation of linear estimators.

**Proposition 9** (Theorem 3.3 in Hayakawa & Suzuki (2020))**.** *Let $\mathcal{F}$ be a class of functions on $\Omega$, and $\mathrm{conv}(\mathcal{F})$ be the convex hull of $\mathcal{F}$ defined as*

$$\mathrm{conv}(\mathcal{F}) := \left\{ \sum_{i=1}^m \alpha_i f_i \mid m \in \mathbb{N}, f_i \in \mathcal{F}, \alpha_i \geq 0, \sum_{i=1}^m \alpha_i = 1 \right\}.$$

*Then, it holds that*

$$\inf_{\widehat{f}: linear} \sup_{f^\circ \in \mathcal{F}} \mathbb{E}_{D_n}\left[ \left\|\widehat{f} - f^\circ\right\|^2_{L^2(P_X)} \right] = \inf_{\widehat{f}: linear} \sup_{f^\circ \in \mathrm{conv}(\mathcal{F})} \mathbb{E}_{D_n}\left[ \left\|\widehat{f} - f^\circ\right\|^2_{L^2(P_X)} \right].$$

This proposition states that the excess risk of estimating the function in $\mathcal{F}$ by linear estimators coincides with that in the convex hull of $\mathcal{F}$. Therefore, if the function class $\mathcal{F}$ is not convex, linear estimators tend to perform poorly since they have to estimate a larger class of functions $\mathrm{conv}(\mathcal{F})$.

Next, we prove the following lemma.

**Lemma 10.** *Let $\eta > 0$ and $\gamma \in (0, \widetilde{s} - 1/p)$. Suppose that $\mathcal{R} := \{R_1, \ldots, R_L\}$ is a family of disjoint hyper-rectangles in $\Omega$ with volume $\lambda(R_l) \simeq \eta^\gamma$ ($l \in [L]$). Then, there is a family of functions $\Psi$ satisfying the following three conditions:*

*(A) For all $\psi \in \Psi$, $\min_{x \in \Omega} \psi(x) = 0$ and $\max_{x \in \Omega} \psi(x) = \eta$;*

*(B) There is a one-to-one correspondence $\psi \leftrightarrow R_m$ between $\Psi$ and $\mathcal{R}$ such that $\mathrm{supp}(\psi) \subseteq R_m$ for the corresponding $R_m$;*

*(C) It holds $\|\psi\|_{L^2(\Omega)} \simeq \eta^{2+\gamma}$ for all $\psi \in \Psi$;*

*(D) For all $\epsilon \in (0, \eta]$ and $\psi \in \Psi$, it holds $\lambda(\{x \mid \eta - \psi(x) \le \epsilon\}) \gtrsim \epsilon^{\gamma}$;*

*(E) It holds $\Psi \subset B_{p,q}^{\boldsymbol{s}}(\Omega)$.*

*Proof.* Let $x_m$ the center of $R_m$. Let $\psi : \Omega \to [0,1]$ be a function in $C^{\infty}$ such that
$$\psi(x) \begin{cases} = 1 + 1/2^{\alpha} - \rho_{\boldsymbol{s},2}(x,0)^{\alpha} & \text{if } \rho_{\boldsymbol{s},2}(x,0) \le 1/2, \\ = 0 & \text{if } \rho_{\boldsymbol{s},2}(x,0) \ge 1, \\ \in (0,1) & \text{otherwise,} \end{cases}$$
where $\alpha := \frac{\overline{s}/\widetilde{s}}{\gamma}$. Let $\psi_l(x) := A \cdot \psi((x - x_l)/\eta)$. We prove that $\Psi := \{\psi_1, \dots, \psi_L\}$ satisfies the desired conditions. Conditions (A) and (B) are obviously satisfied. As for (C), the necessary condition to hold $A - \psi_l(x) \le \epsilon$ is $\rho_{\boldsymbol{s},2}((x - x_l)/\eta, 0)^{\alpha} \le \epsilon/A$. Therefore, we have
$$\lambda(\{x \mid A - \psi_l(x) \le \epsilon\}) \gtrsim \eta^{\overline{s}/\widetilde{s}}(\epsilon/A)^{\overline{s}/(\alpha\widetilde{s})} = \eta^{\overline{s}/\widetilde{s}}(\epsilon/A)^{\gamma} \gtrsim \epsilon^{\gamma},$$
which implies (C). Finally, we prove (D). Since $\psi$ is in $C^{\infty}$, it holds $\psi \in B_{p,q}^{\boldsymbol{s}}(\Omega)$. $\qquad\square$

Finally, we prove Theorem 5.

*Proof of Theorem 5.* Let $\Psi = \{\psi_1, \dots, \psi_L\}$ be the function class given in the above lemma. Let $\{(l_1, l_1'), \dots, (l_{J(k)}, l_{J(k)}')\}$ be the set of pairs of $[L]$ satisfying the following conditions:

(a) $m_i \ne l_i'$ for all $i \in [J(k)]$;

(b) $|\{(l, l') \mid (l, l') \in P, l = l^*\}| \simeq |\{(l, l') \mid (l, l') \in P, l' = l^*\}| \simeq J(k)/M$;

(c) $R_{l_j} \cap \mathrm{supp}\, M_{k,j}^d = R_{l_j'} \cap \mathrm{supp}\, M_{k,j}^d = \emptyset$ for all $j \in [J(k)]$.

Then, let us define the finite function class $\mathcal{F}_0$ as follows:
$$\mathcal{F}_0 := \mathcal{F}_1 \cup \mathcal{F}_2,$$
where
$$\mathcal{F}_1 := \{M_{k,j}^d + \psi_{l_j} - \psi_{l_j'} \mid j \in [J(k)]\}, \quad \mathcal{F}_2 := \{M_{k,j}^d - \psi_{l_j} + \psi_{l_j'} \mid j \in [J(k)]\}.$$
Then, for any $f_1, f_2 \in \mathcal{F}_0$, it holds
$$\|f_1 - f_2\|_{L^2(\Omega)} \le \|f_1\|_{L^2(\Omega)} + \|f_2\|_{L^2(\Omega)} \lesssim \eta^{2+\gamma}.$$
Moreover, we define $\mathcal{F} := \mathrm{conv}(\mathcal{F}_0)$. Since it holds
$$\frac{1}{2}(M_{k,j}^d + \psi_{l_j} - \psi_{l_j'}) + \frac{1}{2}(M_{k,j}^d - \psi_{l_j} + \psi_{l_j'}) = M_{k,j}^d,$$
we have
$$\mathcal{F} \supseteq \{M_{k,j}^d \mid j \in \mathbb{Z}^d, \mathrm{supp}(M_{k,j}^d) \cap \Omega \ne \emptyset\} =: \mathcal{G}.$$
Suzuki & Nitanda (2021) proved in Theorem 5 that it holds
$$\inf_{\widehat{f}:\text{linear}} \sup_{f^{\circ} \in \mathcal{G}} \mathbb{E}_{D_n}\left[\left\|\widehat{f} - f^{\circ}\right\|_{L^2(P_X)}^2\right] \gtrsim n^{-\frac{2\widetilde{s}-v}{2\widetilde{s}+1-v}},$$
where $v := 2(1/p - 1/2)_+$. Therefore, using Proposition 9, we have
$$\inf_{\widehat{f}:\text{linear}} \sup_{f^{\circ} \in \mathcal{F}} \mathbb{E}_{D_n}\left[\left\|\widehat{f} - f^{\circ}\right\|_{L^2(P_X)}^2\right] = \inf_{\widehat{f}:\text{linear}} \sup_{f^{\circ} \in \mathrm{conv}(\mathcal{F})} \mathbb{E}_{D_n}\left[\left\|\widehat{f} - f^{\circ}\right\|_{L^2(P_X)}^2\right]$$
$$\gtrsim \inf_{\widehat{f}:\text{linear}} \sup_{f^{\circ} \in \mathcal{G}} \mathbb{E}_{D_n}\left[\left\|\widehat{f} - f^{\circ}\right\|_{L^2(P_X)}^2\right]$$
$$\gtrsim n^{-\frac{2\widetilde{s}-v}{2\widetilde{s}+1-v}},$$
which completes the proof. $\qquad\square$

## C    Proof of Lemma 8

In this section, we consider a general regression problem for anisotropic Besov spaces. Specifically, we consider $f \in B_{p,q}^{\boldsymbol{s}}$, and let $\widehat{f}$ be an estimator of $f^{\circ}$ defined as

$$\widehat{f} := \operatorname*{arg\,min}_{f \in \Phi(L,W,S,B)} \sum_{i=1}^{n} (y_i - f(x_i))^2, \tag{3}$$

where $x_1, \ldots, x_n$ are i.i.d. samples from a distribution $P_X$, and $y_i = f^{\circ}(x_i) + \xi_i$ with $\xi_i \sim \mathcal{N}(0, \sigma^2)$. We denote $D_n := \{(x_i, y_i)\}_{i=1}^{n}$ as the dataset.

Lemma 8 is directly derived from the following theorem by setting $g(x) = r^*(x) - r^{\circ}(x, y)$.

**Theorem 11** (Localized Estimation Error for Anisotropic Besov Spaces). *Let $f, g \in B_{p,q}^{\boldsymbol{s}}(\Omega)$ with $p, q \in (0, \infty]$, $\boldsymbol{s} \in \mathbb{R}_{>0}^d$, and $\widetilde{s} > 1/p$, suppose that $f(x) \in [-F, F]$ and $g(x) \in [0, T]$ for all $x \in \Omega$ with some $F, T > 0$. Let $\Omega_t := \{x \in \Omega \mid g(x) \leq t\}$ for $t \in [0, T]$ with some $T > 0$. Assume that the following three conditions hold for some constants $C_0, c_0, \mathcal{R} > 0$, and $\beta \in \left[0, \frac{1}{2(\widetilde{s}-1/p)}\right)$:*

*(i) For all $\iota \in (0, c_0]$ and $t \in [0, T]$, it holds $\mathcal{M}(\iota; \Omega_t, \rho_{\boldsymbol{s},2}) \leq C_0\big(1 + \lambda(\Omega_t)\iota^{-\overline{s}/\widetilde{s}}\big)$.*

*(ii) For all $t \in [0, T]$, it holds $\lambda(\Omega_t) \lesssim t^{\beta}$.*

*(iii) It holds $\mathbb{E}_{x \sim P_X}[g(X)] \leq \mathcal{R}$.*

*Let $\varsigma$ be a constant such that $\varsigma \in (0, \widetilde{s} - 1/p)$ for $p < \infty$, and $\varsigma = \widetilde{s}$ for $p = \infty$. Moreover, let $\widehat{f} \in \Phi(L, W, S, B)$ be a estimator defined as (3) with*

$$L \lesssim \log N, \quad W \lesssim N, \quad S \lesssim N \log N, \quad \log B \lesssim \log N,$$

*where $N = n^{\frac{1}{2\widetilde{s}+1}} \mathcal{R}^{\frac{2\beta\varsigma}{2\widetilde{s}+1}}$. If $\mathcal{R}^{-1/\widetilde{s}} < N$, it holds*

$$\mathbb{E}_{D_n}\left[\left\|\widehat{f} - f^{\circ}\right\|_{L^2(P_X)}^2\right] \lesssim \mathcal{R}^{\frac{2\beta\varsigma}{2\widetilde{s}+1}} \cdot n^{-\frac{2\widetilde{s}}{2\widetilde{s}+1}} \log^4(n),$$

*where $\mathbb{E}_{D_n}$ is the expectation with respect to the dataset $D_n$.*

In the rest of this section, we prove Theorem 11.

### C.1    Approximation Error on a Small Set

We first prove the following theorem, which gives the approximation error bound for a fixed small set $\Omega' \subseteq \Omega$.

**Theorem 12** (Approximation Error for Anisotropic Besov Spaces). *Let $\Omega' \subseteq \Omega$ be a measurable set satisfying $\mathcal{M}(\iota; \Omega', \rho_{\boldsymbol{s},2}) \leq C_0\big(1 + \lambda(\Omega')\iota^{-\overline{s}/\widetilde{s}}\big)$ for all $\iota \in (0, c_0]$ with some constants $C_0, c_0 > 0$. Assume that $f \in B_{p,q}^{\boldsymbol{s}}(\Omega)$ with $p, q \in (0, \infty]$, $\boldsymbol{s} \in \mathbb{R}_{>0}^d$, and $\widetilde{s} > \delta_0$, where $\delta_0 := (1/p - 1/r)_+$. Moreover, suppose that $m \in \mathbb{N}$ satisfies $0 < \widetilde{s} < \min\{m, m - 1 + 1/p\}$. Let $\nu \in (0, \frac{\widetilde{s}-\delta_0}{\delta_0})$, and $N \in \mathbb{N}_{>0}$ be a sufficiently large integer. We define $N' := \lambda(\Omega')^{\frac{\nu}{1+\nu}} N$ and $\epsilon := N^{-\widetilde{s}-(1+\nu^{-1})(1/p-\widetilde{s})_+} \log^{-1} N$. Then, there exists an FNN $f \in \Psi(L, W, S, B)$ with*

$$L = L_0, \quad W \lesssim N'W_0, \quad S \lesssim (L-1)W_0^2 N' + N', \quad B = O(N^{d(1+\nu^{-1})(1/p-\widetilde{s})_+}),$$

*such that $\|f - f^{\circ}\|_{L^r(\Omega)} \lesssim N^{-\widetilde{s}}$, where*

$$L_0 := 3 + 2\left\lceil \log_2\left(\frac{3^{d \vee m}}{\epsilon c_{d,m}}\right) + 5\right\rceil \lceil \log_2(d \vee m) \rceil, \quad W_0 := 6dm(m+2) + 2d.$$

*and $c_{d,m}$ is a constant depending only on $d$ and $m$.*

We use the following lemma for the proof of Theorem 12.

**Lemma 13** (Lemma 2 of Suzuki & Nitanda (2021)). *Suppose the function $f \in B_{p,q}^{\boldsymbol{s}}(\Omega)$ and the constants $m \in \mathbb{N}$, $\delta_0$, $\nu$ satisfy the same conditions as in Theorem 12. For an integer $K \in \mathbb{N}$, let $N = \left\lceil 2^{\|K\|_{\boldsymbol{s}'}} \right\rceil$. Moreover, we define $\epsilon$ as the same way as in Theorem 12. Then, there exists $f_N$ such that $\|f - f_N\|_{L^r(\Omega)} \lesssim N^{-\widetilde{s}} \|f\|_{B_{p,q}^{\boldsymbol{s}}}$, and $f_N$ can be written as*

$$f_N(x) = \sum_{(k,j)\in E_N} \alpha_{k,j} M_{k,j}^d(x) := \sum_{k=0}^{K} \sum_{j\in J(k)} \alpha_{k,j} M_{k,j}^d(x) + \sum_{k=K+1}^{K^*} \sum_{i=1}^{n_k} \alpha_{k,j_i} M_{k,j_i}^d(x),$$

*where $K^* = \left\lceil K\left(1 + \frac{1}{\nu}\right) \right\rceil$, $n_k = \left\lceil 2^{\|K\|_{\boldsymbol{s}'} - \nu\left(\|k\|_{\boldsymbol{s}'} - \|K\|_{\boldsymbol{s}'}\right)} \right\rceil$ $(k = K+1, \ldots, K^*)$, $\{j_i\}_{i=1}^{n_k} \subset J(k)$, and the coefficients $(\alpha_{k,j})_{k,j}$ satisfies $\max_{(k,j)\in E_N} |\alpha_{k,j}| \lesssim 2^{K^* \cdot (\underline{s}/\widetilde{s}) \cdot (1/p - \widetilde{s})_+}$.*

We also employ the following lemma to provide the upper-bounds the required number of terms in the decomposition of $f_N$ for approximating on the small set $\Omega'$.

**Lemma 14.** *Let $\iota > 0$, $\boldsymbol{s} = [s_1, \ldots, s_d]^\top \in \mathbb{R}_{>0}^d$ and $A \subset \mathbb{R}^d$ be a compact set. Moreover, let $Q_1, \ldots, Q_N \subseteq \Omega$ be $\rho_{\boldsymbol{s},\infty}$-balls with radius $\iota$. Suppose that $Q_i$'s are pairwise disjoint, and each $Q_i$ intersects with $A$. Then, there exists a constant $C_1 > 0$ such that $N \leq C_1 \cdot \mathcal{M}(\iota; A, \rho_{\boldsymbol{s},2})$, where $C_1$ is a constant that only depends on $d$ and $\boldsymbol{s}$.*

*Proof.* From the definition of the covering number, we can take the cover $\{B(x_j, \iota; \rho_{\boldsymbol{s},2})\}_{j=1}^{m}$ of $A$ with $m = \mathcal{M}(\iota; A, \rho_{\boldsymbol{s},2})$ and $x_1, \ldots, x_m \in A$. Moreover, since $Q_i$'s are $\rho_{\boldsymbol{s},\infty}$-balls with radius $\iota$, for any $x, y \in Q_i$, we have $\left(\max_{j\in[d]} |x_j - y_j|^{s_j/\underline{s}}\right)^{\underline{s}/\overline{s}} \leq \iota$, i.e., $|x_j - y_j|^{s_j/\underline{s}} \leq \iota^{\overline{s}/\underline{s}}$ for all $j \in [d]$.

Since each $Q_i$ have intersection with $A$, there exists $j(i) \in [m]$ such that $Q_i \cap B(x_{j(i)}, \iota; \rho_{\boldsymbol{s},2}) \neq \emptyset$. This implies that we can take $z \in Q_i \cap B(x_{j(i)}, \iota; \rho_{\boldsymbol{s},2})$, and thus for any $y \in Q_i$, we have

$$\rho_{\boldsymbol{s},2}(y, x_{j(i)}) \leq \rho_{\boldsymbol{s},2}(y, z) + \rho_{\boldsymbol{s},2}(z, x_{j(i)})$$

$$\leq \left((y_1 - z_1)^{2s_1/\underline{s}} + \cdots + (y_d - z_d)^{2s_d/\underline{s}}\right)^{\underline{s}/2\overline{s}} + \iota$$

$$\leq \left(d\iota^{2\overline{s}/\underline{s}}\right)^{\underline{s}/2\overline{s}} + \iota = (1 + d^{\underline{s}/2\overline{s}})\iota.$$

Therefore, it holds $Q_i \subseteq B(x_{j(i)}, (1 + d^{\underline{s}/2\overline{s}})\iota; \rho_{\boldsymbol{s},2})$. Taking the union of $i = 1, \ldots, N$, we have

$$\bigcup_{i=1}^{N} Q_i \subseteq \bigcup_{i=1}^{N} B(x_{j(i)}, (1 + d^{\underline{s}/2\overline{s}})\iota; \rho_{\boldsymbol{s},2}) \subseteq \bigcup_{j=1}^{m} B(x_j, (1 + d^{\underline{s}/2\overline{s}})\iota; \rho_{\boldsymbol{s},2}).$$

The volume of the left-most and right-most sets can be evaluated as follows:

$$\lambda\left(\bigcup_{i=1}^{N} Q_i\right) = N \cdot \iota^{\overline{s}/s_1} \cdot \ldots \cdot \iota^{\overline{s}/s_d} = N\iota^{\overline{s}\sum_i \frac{1}{s_i}}$$

$$\lambda\left(\bigcup_{j=1}^{m} B(x_j, (1 + d^{\underline{s}/2\overline{s}})\iota; \rho_{\boldsymbol{s},2})\right) \leq \mathcal{M}(\iota; A, \rho_{\boldsymbol{s},2}) \cdot \lambda(B(x_j, (1 + d^{\underline{s}/2\overline{s}})\iota; \rho_{\boldsymbol{s},2}))$$

$$\lesssim \mathcal{M}(\iota; A, \rho_{\boldsymbol{s},2}) \cdot \iota^{\overline{s}\sum_i \frac{1}{s_i}}.$$

Comparing the two volumes, we have $N \lesssim \mathcal{M}(\iota; A, \rho_{\boldsymbol{s},2})$, which completes the proof. $\qquad \square$

Now, we prove Theorem 12.

*Proof of Theorem 12.* Let $f_N = \sum_{(k,j)\in E_N} \alpha_{k,j} M_{k,j}^d$ be the approximation of $f$ given in Lemma 13. Then, we have $\|f - f_N\|_{L^r(\Omega)} \lesssim N^{-\widetilde{s}} \|f\|_{B_{p,q}^{\boldsymbol{s}}}$. Let $E_N' := \left\{(k,j) \in E_N \mid (k,j) \in E_N, \left(\text{supp } M_{k,j}^d\right) \cap \Omega' \neq \emptyset\right\}$, and

$$E_{N,k}' := \{j \in \mathbb{N} \mid (k,j) \in E_N'\} = \left\{j \in \mathbb{N} \mid (k,j) \in E_N, \left(\text{supp } M_{k,j}^d\right) \cap \Omega' \neq \emptyset\right\},$$

for $k \in [K^*]$. Then, we define $f'_N$ as

$$f'_N = \sum_{(k,j) \in E'_N} \alpha_{k,j} M^d_{k,j} = \sum_{k=0}^{K^*} \sum_{j \in E'_{N,k}} \alpha_{k,j} M^d_{k,j}.$$

By the definition of $E'_N$, we have $f_N(x) = f'_N(x)$ for any $x \in \Omega'$.

Next, we evaluate the number of terms in the decomposition of $f'_N$, i.e., $|E'_N|$. The support of $M^d_{k,j}$ is the hyperrectangle obtained by, for each coordinate, scaling the support $[0, m+1]$ of the one-dimensional B-spline $N_m$ by $2^{\lfloor ks'_i \rfloor}$ and translating it by $j_i$. The $i$-th edge length $e_i$ this hyperrectangle is $(m+1)2^{-\lfloor ks'_i \rfloor} \le 2(m+1)2^{-ks'_i} = 2(m+1)2^{-k\underline{s}/s_i}$. Therefore, for any $x, y$ in the support of $M^d_{k,j}$, we have

$$\rho_{\boldsymbol{s},\infty}(x,y) \le \max_{i=1,\dots,d} |x_i - y_i|^{s_i/\overline{s}} \le \max_{i=1,\dots,d} (2(m+1)2^{-k\underline{s}/s_i})^{s_i/\overline{s}} = 2(m+1)^{\overline{s}/\underline{s}} 2^{-k\underline{s}/\overline{s}}.$$

Therefore, using Lemma 14 with $\iota = 2(m+1)^{\overline{s}/\underline{s}} 2^{-k\underline{s}/\overline{s}}$, we have we have

$$\left| E'_{N,k} \right| \lesssim \textcolor{red}{\mathcal{M}(2(m+1)^{\overline{s}/\underline{s}} 2^{-k\underline{s}/\overline{s}}; \Omega', \rho_{\boldsymbol{s},2})}$$

$$\lesssim \lambda(\Omega') \left[ (2^{-k\underline{s}/\overline{s}})^{\overline{s}/\widetilde{s}} \right]^{-1} = \lambda(\Omega') 2^{k\underline{s}/\widetilde{s}}.$$

Moreover, for $k \ge K$, we have $\left| E'_{N,k} \right| \le n_k \lesssim 2^{\|K\|_{\boldsymbol{s}'} - \nu(\|k\|_{\boldsymbol{s}'} - \|K\|_{\boldsymbol{s}'})}$. Hence, for any $K^\circ \ge K$, we have

$$|E'_N| \lesssim \sum_{k=0}^{K^\circ} \left( 1 + \lambda(\Omega') 2^{k\underline{s}/\widetilde{s}} \right) + \sum_{k=K^\circ+1}^{K^*} 2^{\|K\|_{\boldsymbol{s}'} - \nu(\|k\|_{\boldsymbol{s}'} - \|K\|_{\boldsymbol{s}'})}.$$

Let us dermine $K^\circ$ to make the right-hand side the minimum. Since $1 + \lambda(\Omega') 2^{dk}$ is increasing, and $2^{\|K\|_{\boldsymbol{s}'} - \nu(\|k\|_{\boldsymbol{s}'} - \|K\|_{\boldsymbol{s}'})}$ is decreasing with respect to $k$, for the best choice of $K^\circ$, we have

$$\lambda(\Omega') 2^{K^\circ \underline{s}/\widetilde{s}} \simeq 2^{\|K\|_{\boldsymbol{s}'} - \nu(\|K^\circ\|_{\boldsymbol{s}'} - \|K\|_{\boldsymbol{s}'})}.$$

The right-hand side equals to $2^{K\underline{s}/\widetilde{s} - \nu(K^\circ \underline{s}/\widetilde{s} - K\underline{s}/\widetilde{s})}$ up to a constant factor. Therefore, we have

$$2^{-(1+\nu)(K^\circ - K) \cdot \underline{s}/\widetilde{s}} \simeq \lambda(\Omega'),$$

which implies $2^{(K^\circ - K) \cdot \underline{s}/\widetilde{s}} \simeq \lambda(\Omega')^{-\frac{1}{1+\nu}}$. For $K^\circ$ satisfying this condition, we have

$$|E'_N| \lesssim \lambda(\Omega') \frac{2^{K^\circ \underline{s}/\widetilde{s}}}{1 - 2^{\underline{s}/\widetilde{s}}} + 2^{(K - \nu(K^\circ - K)) \cdot \underline{s}/\widetilde{s}} \frac{1}{1 - 2^{-\nu\underline{s}/\widetilde{s}}}$$

$$\lesssim \lambda(\Omega') \cdot 2^{K\underline{s}/\widetilde{s}} \lambda(\Omega')^{-\frac{1}{1+\nu}} + 2^{K\underline{s}/\widetilde{s}} \lambda(\Omega')^{\frac{\nu}{1+\nu}}$$

$$\lesssim \lambda(\Omega')^{\frac{\nu}{1+\nu}} 2^{\|K\|_{\boldsymbol{s}'}} = \lambda(\Omega')^{\frac{\nu}{1+\nu}} \cdot N.$$

The remaining part of the proof is adapted from the proof of Proposition 2 of Suzuki & Nitanda (2021). Specifically, from Lemma 1 of Suzuki (2018), for all $k$ and $j$, there exists an FNN $\widehat{M}_{k,j}$ such that $\left\| \widehat{M}^d_{k,j} - M^d_{k,j} \right\|_{L^\infty(\mathbb{R}^d)} \le \epsilon$, and $\widehat{M}^d_{k,j} = 0$ in $x \notin [0, m+1]^d$. Using these networks, we can construct $\hat{f} \in \Psi(L, W, S, B)$ with $L, W, S, B$ as in the statement of the theorem such that

$$\hat{f}(x) = \sum_{(k,j) \in E'_N} \alpha_{k,j} \widehat{M}^d_{k,j}(x).$$

Then, we have

$$\left| f'_N(x) - \hat{f}(x) \right| \le \sum_{(k,j) \in E'_N} |\alpha_{k,j}| \cdot \left| M^d_{k,j}(x) - \widehat{M}^d_{k,j}(x) \right|$$

$$\le \epsilon \sum_{(k,j) \in E'_N} |\alpha_{k,j}| \cdot \mathbb{1}_{\operatorname{supp} M^d_{k,j}}(x).$$

For each $x \in \Omega$, the number of $(k, j)$ such that $x \in \text{supp } M_{k,j}^d$ is at most $(m+1)^d (1 + K^*)$. Combining with the upper-bound $\max_{(k,j) \in E'_N} |\alpha_{k,j}|$ given in Lemma 13, Therefore, we have

$$
\left| f'_N(x) - \hat{f}(x) \right| \leq \epsilon \max_{(k,j) \in E'_N} |\alpha_{k,j}| \cdot (m+1)^d (1 + K^*)
$$
$$
\lesssim \epsilon 2^{K^* \cdot (\underline{s}/\widetilde{s}) \cdot (1/p - \widetilde{s})_+} (1 + K^*).
$$

Since it holds

$$
2^{K^* \cdot (\underline{s}/\widetilde{s})} \simeq 2^{K(\underline{s}/\widetilde{s}) \cdot (1 + \nu^{-1})} \simeq 2^{\sum_{j=1}^d \lfloor K\underline{s}/s_j \rfloor \cdot (1 + \nu^{-1})} = 2^{\|K\|_{s'} \cdot (1 + \nu^{-1})} \simeq N^{1 + \nu^{-1}},
$$

we have

$$
\left| f'_N(x) - \hat{f}(x) \right| \lesssim \epsilon N^{(1 + \nu^{-1})(1/p - \widetilde{s})_+} \log N \leq N^{-\widetilde{s}}.
$$

Moreover, the absolute values of parameters used in $\widehat{M}_{k,j}^d$ is at most $2^{K^*} \lesssim N^{d(1+\nu^{-1})(1/p-\widetilde{s})_+}$, which completes the proof. $\square$

## C.2 Localized Approximation Error Bound

Next, we prove the following theorem, which considers the family of sublevel sets of $g$ as in Theorem 11.

**Theorem 15** (Localized Approximation Error for Anisotropic Besov Spaces). *Suppose that the functions $f$, $g$, the family of sets $\{\Omega_t\}_{t \in [0,T]}$, and the constants $C_0, c_0, \mathcal{R}, \beta, \varsigma$ satisfy the same conditions as in Theorem 11. Moreover, suppose that $m \in \mathbb{N}$ satisfies $0 < \widetilde{s} < \min\{m, m - 1 + 1/p\}$. Let $N \in \mathbb{R}_{>0}$ be a sufficiently large real number. Then, there exists an FNN $f \in \Psi(L, W, S, B)$ with*

$$
L \lesssim \log N + \log \mathcal{R}^{-1}, \quad W \lesssim N \log \mathcal{R}^{-1} + \mathcal{R}^{-1/\widetilde{s}},
$$
$$
S \lesssim N \log N \log \mathcal{R}^{-1} + \mathcal{R}^{-1/\widetilde{s}} \log \mathcal{R}^{-1}, \quad \log B \lesssim \log \mathcal{R}^{-1},
$$

*such that $\|f - f^\circ\|_{L^2(\Omega)} \lesssim N^{-\widetilde{s}} \mathcal{R}^{\beta(\widetilde{s} - 1/p)}$.*

*Proof.* Applying Theorem 12 with $r = \infty$, we have that, for all $t \in [0, T]$, there exists an FNN $f'_t \in \Psi(L', W'_t, S'_t, B')$ with

$$
L' \lesssim \log(\epsilon^{-1}) \lesssim \log\left(N^{\widetilde{s}} \log N\right) \lesssim \log N' + \log t^{-1},
$$
$$
W'_t \lesssim N' W_0 \lesssim N',
$$
$$
S'_t \lesssim N' \log N + N' \lesssim N'(\log N' + \log t^{-1}),
$$
$$
B' \lesssim 1,
$$

such that

$$
\sup_{x \in \Omega_t} |f'_t(x) - f^\circ(x)| \lesssim N^{-\widetilde{s}} \lesssim (N')^{-\widetilde{s}} t^{\beta \cdot \frac{\nu \widetilde{s}}{1 + \nu}} \lesssim (N')^{-\widetilde{s}} t^{\beta \varsigma}
$$

Let $a_{-1} = 0$ and $a_i = 2^i \mathcal{R}$ for $i = 0, \ldots, I$ with $I := \lceil \log_2(2F/\mathcal{R}) \rceil$. Then, for $i = 0, \ldots, I$ and any $N \in \mathbb{R}$, we can construct an FNN $f_i \in \Psi(L_i, W_i, S_i, B_i)$ with

$$
L_i \lesssim \log N + \log \mathcal{R}^{-1}, \quad W_i \lesssim N, \quad S_i \lesssim N(\log N + \log \mathcal{R}^{-1}), \quad B_i \lesssim 1,
$$

such that

$$
\sup_{x \in \Omega_{a_i}} |f_i(x) - f^\circ(x)| \lesssim N^{-\widetilde{s}} (2^i \mathcal{R})^{\beta \varsigma}.
$$

Moreover, applying Theorem 12 for $g$ with $N \leftarrow \mathcal{R}^{-1/\widetilde{s}}$ and $\mathcal{R} \leftarrow T$, we have an FNN $\widetilde{g} \in \Psi(L, W_g, S_g, B_g)$ with

$$
L_g \lesssim \log \mathcal{R}^{-1}, \quad W_g \lesssim \mathcal{R}^{-1/\widetilde{s}}, \quad S_g \lesssim \mathcal{R}^{-1/\widetilde{s}} \log \mathcal{R}^{-1}, \quad \log B_g \lesssim \log \mathcal{R}^{-1},
$$

such that $\sup_{x \in \Omega} |g(x) - \widetilde{g}(x)| \lesssim \mathcal{R}/8$.

For $i = 0, \ldots, I$, we can construct an FNN $\phi_i \in \Phi(L, W, S, B)$ with $L, W, S \lesssim 1$ and $\log B \lesssim \log \mathcal{R}^{-1}$ such that

$$
\phi_i(x) = \begin{cases}
0 & (x \leq a_{i-1} - \mathcal{R}/4), \\
(x - a_{i-1} + \eta)/(2\eta) & (a_{i-1} - \mathcal{R}/4 < x < a_{i-1} - \mathcal{R}/8), \\
1 & (a_{i-1} + \eta \leq x \leq a_i - \mathcal{R}/4), \\
(a_i + \eta - x)/(2\eta) & (a_i - \mathcal{R}/4 < x < a_i - \mathcal{R}/8), \\
0 & (a_i - \mathcal{R}/8 \leq x).
\end{cases}
$$

Then, we have $\sum_{i=0}^{I} \phi_i(x) = 1$ for all $x \in [0, 2F]$. Moreover, since $\phi_i(x) > 0$ only if $x \in [a_{i-1} - \mathcal{R}/4, a_i - \mathcal{R}/8]$, the necessary condition to $\phi_i(\widetilde{g}(x)) > 0$ is $g(x) \in [a_{i-1} - 3\mathcal{R}/8, a_i]$.

Now, we define $\check{f}$ as

$$
\check{f}(x) := \sum_{i=0}^{I} \phi_i(\widetilde{g}(x)) f_i(x).
$$

Let us consider $x \in \Omega$ such that $g(x) \in [a_{i-1}, a_i]$ for some $i \in [0, I]$. Then, we have $g(x) \in [a_{i-1} - 3\mathcal{R}/8, a_i]$, which implies then $\phi_j(\widetilde{g}(x)) = 0$ for $j \neq i, i-1$. Therefore, we have

$$
\left| \check{f}(x) - f^\circ(x) \right| \leq \phi_i(\widetilde{g}(x)) |f_i(x) - f^\circ(x)| + \phi_{i-1}(\widetilde{g}(x)) \left| f_{a_{i-1}}(x) - f^\circ(x) \right|
$$

$$
\leq \max\{ |f_i(x) - f^\circ(x)|, \left| f_{a_{i-1}}(x) - f^\circ(x) \right| \}
$$

$$
\lesssim N^{-\widetilde{s}} (2^i \mathcal{R})^{\beta\varsigma}.
$$

Moreover, for $x \sim P_X$, the probability of $x \in [a_{i-1}, a_i]$ can be bounded as

$$
\mathbb{P}_{x \sim P_X}[g(x) \in [a_{i-1}, a_i]] \leq \mathbb{P}_{x \sim P_X}[g(x) \geq 2^{i-1}\mathcal{R}] \leq \frac{\mathbb{E}_{x \sim P_X}[g(x)]}{2^{i-1}\mathcal{R}} \leq 2^{-(i-1)}.
$$

Therefore, we have

$$
\left\| \check{f} - f^\circ \right\|_{L^2(P_X)}^2 = \mathbb{E}_{x \sim P_X}\left[ \left| \check{f}(x) - f^\circ(x) \right|^2 \right]
$$

$$
\lesssim \sum_{i=0}^{I} N^{-2\widetilde{s}} (2^i \mathcal{R})^{2\beta\varsigma} \mathbb{P}_{x \sim P_X}[g(x) \in [a_{i-1}, a_i]]
$$

$$
\lesssim N^{-2\widetilde{s}} \mathcal{R}^{2\beta\varsigma} \sum_{i=0}^{I} (2^{2\beta\varsigma - 1})^i
$$

$$
\lesssim N^{-2\widetilde{s}} \mathcal{R}^{2\beta\varsigma}.
$$

Finally, since $\phi(\widetilde{g}(x))$ and $f_i(x)$ are bounded by constants for all $x \in \Omega$, Lemma 23 implies that there exists an FNN $f \in \Psi(L, W, S, B)$ with

$$
L \lesssim \log N + \log \mathcal{R}^{-1}, \quad W \lesssim N \log \mathcal{R}^{-1} + \mathcal{R}^{-1/\widetilde{s}},
$$

$$
S \lesssim N \log N \log \mathcal{R}^{-1} + \mathcal{R}^{-1/\widetilde{s}} \log \mathcal{R}^{-1}, \quad \log B \lesssim \log \mathcal{R}^{-1},
$$

such that $\|f - \check{f}\|_\infty \leq N^{-\widetilde{s}} \mathcal{R}^{\beta\varsigma}$. Then, we have

$$
\|f - f^\circ\|_{L^2(P_X)} \leq \|f - \check{f}\|_{L^2(P_X)} + \|\check{f} - f^\circ\|_{L^2(P_X)} \lesssim N^{-\widetilde{s}} \mathcal{R}^{\beta\varsigma},
$$

which completes the proof. $\qquad\square$

## C.3 Proof of Theorem 11

Finally, we prove Theorem 11.

We utilize the following proposition for the proof.

**Proposition 16** (Schmidt-Hieber (2020); Hayakawa & Suzuki (2020)). *Let $\mathcal{F}$ be a set of functions. Let $\widehat{f}$ be the least-squares estimator in $\mathcal{F}$:*

$$
\widehat{f} := \underset{f \in \mathcal{F}}{\arg\min} \sum_{i=1}^{n} (y_i - f(x_i))^2,
$$

*Assume that $\|f^\circ\|_\infty \leq F$ and $\|f\|_\infty \leq F$ for all $f \in \mathcal{F}$. If $\delta > 0$ satisfies $\mathcal{M}(\delta; \mathcal{F}, \|\cdot\|_\infty) \geq 3$, then it holds that*

$$\mathbb{E}_{D_n}\left[\left\|\widehat{f} - f^\circ\right\|_{L^2(P_X)}^2\right] \lesssim C\left[\inf_{f \in \mathcal{F}}\|f - f^\circ\|_{L^2(P_X)}^2 + (F^2 + \sigma^2)\frac{\log \mathcal{M}(\delta; \mathcal{F}, \|\cdot\|_\infty)}{n} + \delta(F + \sigma)\right],$$

*where $C > 0$ is a universal constant.*

To upper bound the covering number of the function class of FNNs, we use the following result.

**Lemma 17** (Lemma 6 of Suzuki & Nitanda (2021)). *The covering number of $\Phi(L, W, S, B)$ can be bounded as*

$$\log \mathcal{M}(\delta; \Phi(L, W, S, B), \|\cdot\|_\infty) \leq 2SL \log((B+1)(W+1)) + S \log\left(\delta^{-1}L\right),$$

Now, we prove Theorem 11.

*Proof of Theorem 11.* The proof is basically the same as that of Theorem 2 of Suzuki & Nitanda (2021). The difference is that our proof explicitly provides the dependency on $\mathcal{R}$.

In the following, we assume that $N \geq \mathcal{R}^{-1/\widetilde{s}}$. Then, the configuration of $L, W, S, B$ in Theorem 15 can be simplified as

$$L \lesssim \log N, \quad W \lesssim N \log N, \quad S \lesssim N \log^2 N, \quad \log B \lesssim \log N.$$

Let $\mathcal{F} := \Phi(L, W, S, B)$. Then, the covering number of the function class $\mathcal{F}$ can be bounded as

$$\log \mathcal{N}(\delta, \mathcal{F}, \|\cdot\|_\infty) \lesssim N \log^3 N(\log N + \log\log N) + N \log^2 N(\log(\delta^{-1}) + \log\log N)$$
$$\lesssim N \log^2 N(\log^2 N + \log(\delta^{-1})),$$

Using Proposition 16 and setting $\delta := 1/n$, estimation error can be bounded as

$$\mathbb{E}_{D_n}\left[\left\|\widehat{f} - f^\circ\right\|_{L^2(P_X)}^2\right] \lesssim \left\|\widehat{f} - f^\circ\right\|_{L^\infty(\text{supp}(P_X))}^2 + \frac{N \log^2 N(\log^2 N + \log(\delta^{-1}))}{n} + \frac{1}{n}$$
$$\lesssim N^{-2\widetilde{s}}\mathcal{R}^{2\beta\varsigma} + \frac{N \log^2 N(\log^2 N + \log n)}{n} + \frac{1}{n}.$$

Let us set $N = n^{\frac{1}{2\widetilde{s}+1}} \mathcal{R}^{\frac{2\beta\varsigma}{2\widetilde{s}+1}}$. Then, if $N \geq \mathcal{R}^{-1/\widetilde{s}}$, we have

$$\mathbb{E}_{D_n}\left[\left\|\widehat{f} - f^\circ\right\|_{L^2(P_X)}^2\right] \lesssim n^{-\frac{2\widetilde{s}}{2\widetilde{s}+1}} \mathcal{R}^{\frac{2\beta\varsigma}{2\widetilde{s}+1}} \log^4(n),$$

which completes the proof. $\qquad\square$

## D  PROOF OF THE REGRET BOUND

For the convenience of the discussion below, we define $\mathcal{C}(x; \pi_1, \pi_2)$ and $\mathcal{C}(\pi_1, \pi_2)$ for two policies $\pi_1, \pi_2$ as

$$\mathcal{C}(x; \pi_1, \pi_2) := \mathbb{E}_{y \sim \pi_1(\cdot|x)}\left[\frac{\pi_1(y \mid x)}{\pi_2(y \mid x)}\right], \quad \mathcal{C}(\pi_1, \pi_2) := \mathbb{E}_{x \sim P_X}[\mathcal{C}(x; \pi_1, \pi_2)].$$

The value $\mathcal{C}(x; \pi_1, \pi_2)$ is referred to as the *coverage* in Huang et al. (2025a). This value is known to play an important role in the regret analysis of inference-time alignment. Specifically, this value quantifies how well the policy $\pi_{\text{ref}}$ induced by the pre-trained model captures the comparator policy $\pi^*$.

### D.1  PREPARATIONS: PROPERTIES OF `InferenceTimePessimism`

For $\mu > 0$, we define $\pi_\mu^\chi$ by

$$\pi_\mu^\chi(\cdot \mid x) := \underset{\pi:\text{density on }\Omega_Y}{\arg\max} \; \mathbb{E}_{y \sim \pi}[\widehat{r}(x, y)] - \mu \cdot \chi^2(\pi \,\|\, \pi_{\text{ref}}(\cdot \mid x)).$$

Then, we can write $\pi_\mu^\chi$ in a closed form as

$$\pi_\mu^\chi(y \mid x) = \pi_{\mathrm{ref}}(y \mid x)\big[\mu^{-1}(\widehat{r}(x,y) - \theta_\mu)\big]_+,$$

where $\theta_\mu$ is the normalizing constant such that $\int \pi_\mu^\chi(y \mid x)\,\mathrm{d}y = 1$. `InferenceTimePessimism` is a practical algorithm to get samples from $\pi_{\mu,N}^{\mathtt{Pes}}$ that approximates $\pi_\mu^\chi$, where $N \in \mathbb{Z}_{>0}$ is the number of samples to be drawn from $\pi_{\mathrm{ref}}(\cdot \mid x)$.

Now, we present the regret bound of `InferenceTimePessimism` in Huang et al. (2025a).

**Proposition 18** (Theorem 4.1 in Huang et al. (2025a))**.** *Let $\widehat{r}$ be an arbitrary estimator of reward $r^\circ$, and we define $\epsilon_{\mathtt{RM}}^2(x) := \mathbb{E}_{y\sim\pi_{\mathrm{ref}}(\cdot|x)}[(\widehat{r}(x,y) - r^\circ(x,y))^2]$. Moreover, let $\pi^*$ be a comparator policy, Then,* `InferenceTimePessimism` *satisfies*

$$\mathbb{E}_{y\sim\pi^*}[r^\circ(x,y)] - \mathbb{E}_{y\sim\pi_{\mu,N}^{\mathtt{Pes}}(\cdot|x)}[r^\circ(x,y)]$$

$$\lesssim \sqrt{\mathcal{C}(x;\pi^*,\pi_{\mathrm{ref}})\cdot\epsilon_{\mathtt{RM}}^2(x)} + \mu\cdot\mathcal{C}(x;\pi^*,\pi_{\mathrm{ref}}) + \mu^{-1}\cdot\epsilon_{\mathtt{RM}}^2(x) + \mu^{-1}\cdot\epsilon_{\mathtt{RM}}(x)\exp\left(-\frac{\mu N}{C_1(R+\mu)}\right),$$

*for some constant $C_1 > 0$. Setting $\mu \simeq \sqrt{\frac{\epsilon_{\mathtt{RM}}^2(x)}{\mathcal{C}(x;\pi^*,\pi_{\mathrm{ref}})}}$ and $N \gtrsim \widetilde{\Omega}\Big(\sqrt{\frac{\mathcal{C}(x;\pi^*,\pi_{\mathrm{ref}})}{\epsilon_{\mathtt{RM}}^2(x)}}\Big)$, it holds*

$$\mathbb{E}_{y\sim\pi^*}[r^\circ(x,y)] - \mathbb{E}_{y\sim\pi_{\mu,N}^{\mathtt{Pes}}(\cdot|x)}[r^\circ(x,y)] \lesssim \sqrt{\mathcal{C}(x;\pi^*,\pi_{\mathrm{ref}})\cdot\epsilon_{\mathtt{RM}}^2(x)}.$$

### D.2 ANALYSIS FOR THE FIRST STEP

In this section, we analyze the regret of `InferenceTimePessimism` when the reward function $r^\circ$ belongs to the anisotropic Besov space $B_{p,q}^{\boldsymbol{s}}(\Omega)$.

We first prove the following lemma, which is important to connect **(A1)** of Assumption 6 and Proposition 18.

**Lemma 19.** *Suppose that $r^\circ \in B_{p,q}^{\boldsymbol{s}}(\Omega)$ satisfies (A1) of Assumption 6. Then, for any $\epsilon \in (0,\epsilon_0]$, there exists a comparator policy $\pi_\epsilon^*$ satisfying the following two conditions:*

$$\text{(i) } \mathbb{E}_{x\sim P_X}\big[r^*(x) - \mathbb{E}_{y\sim\pi_\epsilon^*(\cdot|x)}[r^\circ(x,y)]\big] \leq \epsilon, \qquad \text{(ii) } \mathcal{C}(\pi_\epsilon^*,\pi_{\mathrm{ref}}) \leq \epsilon^{-\gamma}.$$

*Proof.* If we set the policy $\pi_\epsilon^*$ as

$$\pi_\epsilon^*(y \mid x) := \frac{\mathbb{1}_{S_\epsilon(x)}(y)}{\lambda(S_\epsilon(x))}$$

then the two conditions are satisfied. Indeed, the condition (i) is satisfied since

$$\mathbb{E}_{x\sim P_X}\big[r^*(x) - \mathbb{E}_{y\sim\pi_\epsilon^*(\cdot|x)}[r^\circ(x,y)]\big] = \mathbb{E}_{x\sim P_X}\big[\mathbb{E}_{y\sim\pi_\epsilon^*(\cdot|x)}[r^*(x) - r^\circ(x,y)]\big]$$

$$\leq \mathbb{E}_{x\sim P_X}\left[\epsilon\cdot\frac{1}{\lambda(S_\epsilon(x))}\cdot\lambda(S_\epsilon(x))\right]$$

$$= \epsilon,$$

and the condition (ii) is satisfied since

$$\mathcal{C}(\pi_\epsilon^*,\pi_{\mathrm{ref}}) = \mathbb{E}_{x\sim P_X}\left[\mathbb{E}_{y\sim\pi_\epsilon^*(\cdot|x)}\left[\frac{\pi_\epsilon^*(y \mid x)}{\pi_{\mathrm{ref}}(y \mid x)}\right]\right] = \mathbb{E}_{x\sim P_X}\left[\int\frac{\pi_\epsilon^*(y \mid x)^2}{\pi_{\mathrm{ref}}(y \mid x)}\,\mathrm{d}y\right]$$

$$\leq \mathbb{E}_{x\sim P_X}\left[\int\frac{1/\lambda(S_\epsilon(x))^2}{\underbrace{\pi_{\min}}}\,\mathrm{d}y\right] \lesssim \mathbb{E}_{x\sim P_X}\left[\frac{1}{\lambda(S_\epsilon(x))}\right] \lesssim \epsilon^{-\gamma}.$$

This completes the proof. $\qquad\square$

In this subsection, the reward model $\widehat{r}$ is trained with $n$ samples drawn from $P_X \otimes \pi_{\mathrm{ref}}(\cdot \mid x)$. The true reward value are queried from the oracle, thus we obtain $n$ samples $\{(x_i,y_i,r^\circ(x_i,y_i))\}_{i=1}^n$. Using these samples, we can construct an estimator $\widehat{r} \in \Phi(L,W,S,B)$ of $r^\circ$ satisfying

$$\epsilon_{\mathtt{RM}} := \big(\mathbb{E}_{x\sim P_X}[\epsilon_{\mathtt{RM}}(x)^2]\big)^{1/2} = \|\widehat{r} - r^\circ\|_{L^2(P_X\otimes\pi_{\mathrm{ref}})} \lesssim n^{-\frac{\widetilde{s}}{2\widetilde{s}+1}}.$$

**Theorem 20.** *Let $r^\circ \in B_{p,q}^{\boldsymbol{s}}(\Omega)$ with $\boldsymbol{s} \in \mathbb{R}_{>0}^d$, $p,q \in [1,\infty]$, $\widetilde{s} \geq 1/p$. Suppose that $n$ oracles can be used during training. Under Assumption 6, $\pi_{\mu,N}^{\mathtt{Pes}}$ achieves*

$$\mathbb{E}_{x \sim P_X}[r^*(x) - \mathbb{E}_{y \sim \pi_{\mu,N}^{\mathtt{Pes}}(\cdot|x)}[r^\circ(x,y)]] \lesssim \epsilon_{\mathtt{RM}}^{\frac{2}{2+\gamma}} \lesssim n^{-\frac{2}{2+\gamma} \cdot \frac{\widetilde{s}}{2\widetilde{s}+1}},$$

*for $\mu \simeq n^{-\frac{\widetilde{s}}{2\widetilde{s}+1} \cdot \frac{2(1+\gamma)}{2+\gamma}}$ and $N \gtrsim n^{\frac{\widetilde{s}}{2\widetilde{s}+1} \cdot \frac{2(1+\gamma)}{2+\gamma}} \log(n)$.*

*Proof.* Let $\pi_\epsilon^*$ is the policy satisfying the conditions of Lemma 19 for an arbitrary $\epsilon > 0$. Then, Proposition 18 implies that

$$\mathbb{E}_{y \sim \pi_\epsilon^*}[r^\circ(x,y)] - \mathbb{E}_{y \sim \pi_{\mu,N}^{\mathtt{Pes}}(\cdot|x)}[r^\circ(x,y)]$$

$$\lesssim \mathcal{C}(x;\pi_\epsilon^*,\pi_{\mathrm{ref}})^{1/2}\epsilon_{\mathtt{RM}}(x) + \mu \cdot \mathcal{C}(x;\pi_\epsilon^*,\pi_{\mathrm{ref}}) + \mu^{-1} \cdot \epsilon_{\mathtt{RM}}^2(x) + \mu^{-1} \cdot \epsilon_{\mathtt{RM}}(x)\exp\left(-\frac{\mu N}{C_1(R+\mu)}\right)$$

Taking the expectation over $x \sim P_X$ and using Cauthy-Schwarz inequality, we have

$$\mathbb{E}_{x \sim P_X}\left[\mathbb{E}_{y \sim \pi_\epsilon^*}[r^\circ(x,y)] - \mathbb{E}_{y \sim \pi_{\mu,N}^{\mathtt{Pes}}(\cdot|x)}[r^\circ(x,y)]\right]$$

$$\lesssim \left(\mathbb{E}_{x \sim P_X}[\mathcal{C}(x;\pi_\epsilon^*,\pi_{\mathrm{ref}})]\right)^{1/2} \cdot \left(\mathbb{E}_{x \sim P_X}[\epsilon_{\mathtt{RM}}^2(x)]\right)^{1/2} + \mu \cdot \mathbb{E}_{x \sim P_X}[\mathcal{C}(x;\pi_\epsilon^*,\pi_{\mathrm{ref}})]$$

$$+ \mu^{-1} \cdot \mathbb{E}_{x \sim P_X}[\epsilon_{\mathtt{RM}}^2(x)] + \mu^{-1} \cdot \mathbb{E}_{x \sim P_X}[\epsilon_{\mathtt{RM}}(x)] \cdot \exp\left(-\frac{\mu N}{C_1(R+\mu)}\right)$$

$$\lesssim \left(\mathcal{C}(\pi_\epsilon^*,\pi_{\mathrm{ref}})\right)^{1/2} \cdot \left(\mathbb{E}_{x \sim P_X}[\epsilon_{\mathtt{RM}}^2(x)]\right)^{1/2} + \mu \cdot \mathcal{C}(\pi_\epsilon^*,\pi_{\mathrm{ref}})$$

$$+ \mu^{-1} \cdot \mathbb{E}_{x \sim P_X}[\epsilon_{\mathtt{RM}}^2(x)] + \mu^{-1} \cdot \left(\mathbb{E}_{x \sim P_X}[\epsilon_{\mathtt{RM}}^2(x)]\right)^{1/2} \cdot \exp\left(-\frac{\mu N}{C_1(R+\mu)}\right).$$

If we set

$$\mu \simeq \mathcal{C}(\pi_\epsilon^*,\pi_{\mathrm{ref}})^{-1/2} \cdot \left(\mathbb{E}_{x \sim P_X}[\epsilon_{\mathtt{RM}}^2(x)]\right)^{1/2}, \quad N \gtrsim \mu^{-1}\log\left(\mathbb{E}_{x \sim P_X}[\epsilon_{\mathtt{RM}}^2(x)]\right)$$

then we have

$$\mathbb{E}_{x \sim P_X}\left[\mathbb{E}_{y \sim \pi_\epsilon^*}[r^\circ(x,y)] - \mathbb{E}_{y \sim \pi_{\mu,N}^{\mathtt{Pes}}(\cdot|x)}[r^\circ(x,y)]\right] \lesssim \left(\mathcal{C}(\pi_\epsilon^*,\pi_{\mathrm{ref}})\right)^{1/2} \cdot \left(\mathbb{E}_{x \sim P_X}[\epsilon_{\mathtt{RM}}^2(x)]\right)^{1/2}.$$

Using the property of $\pi_\epsilon^*$ in Lemma 19 (ii) and the error bound of $\left(\mathbb{E}_{x \sim P_X}[\epsilon_{\mathtt{RM}}^2(x)]\right)^{1/2}$, we have

$$\mathbb{E}_{x \sim P_X}\left[\mathbb{E}_{y \sim \pi_\epsilon^*}[r^\circ(x,y)] - \mathbb{E}_{y \sim \pi_{\mu,N}^{\mathtt{Pes}}(\cdot|x)}[r^\circ(x,y)]\right] \lesssim \epsilon^{-\frac{\gamma}{2}} \cdot n^{-\frac{\widetilde{s}}{2\widetilde{s}+1}}.$$

Combining this and the property of $\pi_\epsilon^*$ in Lemma 19 (i), we have

$$\mathbb{E}_{x \sim P_X}\left[r^*(x) - \mathbb{E}_{y \sim \pi_{\mu,N}^{\mathtt{Pes}}(\cdot|x)}[r^\circ(x,y)]\right]$$

$$\leq \mathbb{E}_{x \sim P_X}\left[r^*(x) - \mathbb{E}_{y \sim \pi_\epsilon^*}[r^\circ(x,y)]\right] + \mathbb{E}_{x \sim P_X}\left[\mathbb{E}_{y \sim \pi_\epsilon^*}[r^\circ(x,y)] - \mathbb{E}_{y \sim \pi_{\mu,N}^{\mathtt{Pes}}}[r^\circ(x,y)]\right]$$

$$\lesssim \epsilon + \epsilon^{-\frac{\gamma}{2}} \cdot n^{-\frac{\widetilde{s}}{2\widetilde{s}+1}}.$$

The right-hand side is minimized when $\epsilon \simeq n^{-\frac{2}{2+\gamma} \cdot \frac{\widetilde{s}}{2\widetilde{s}+1}}$. Thus, we have

$$\mathbb{E}_{x \sim P_X}\left[r^*(x) - \mathbb{E}_{y \sim \pi_{\mu,N}^{\mathtt{Pes}}(\cdot|x)}[r^\circ(x,y)]\right] \lesssim n^{-\frac{2}{2+\gamma} \cdot \frac{\widetilde{s}}{2\widetilde{s}+1}}.$$

Moreover, we have

$$\mu \simeq n^{\frac{\widetilde{s}}{2\widetilde{s}+1} \cdot \frac{\gamma}{2+\gamma}} \cdot n^{-\frac{\widetilde{s}}{2\widetilde{s}+1}} = n^{-\frac{\widetilde{s}}{2\widetilde{s}+1} \cdot \frac{2(1+\gamma)}{2+\gamma}}, \quad N \gtrsim n^{\frac{\widetilde{s}}{2\widetilde{s}+1} \cdot \frac{2(1+\gamma)}{2+\gamma}}\log(n).$$

$\square$

## D.3  IMPROVED REGRET VIA MULTI-STEP TRAINING

We now analyze the multi-step training algorithm described in Algorithm 2. First, we prove the following lemma, which corresponds to Lemma 19 in single-step analysis.

**Lemma 21.** *Suppose that $r^\circ \in B_{p,q}^s(\Omega)$ satisfies **(A1)** of Assumption 6. Moreover, let $\widehat{\pi}$ be a policy satisfying $\mathbb{E}_{x \sim P_X} \mathbb{E}_{y \sim \widehat{\pi}(\cdot|x)}[r^*(x) - r^\circ(x,y)] \leq \mathcal{R}$. Additionally, let $\check{\pi}(\cdot \mid x)$ is a distribution of $y + z$ where $y \sim \widehat{\pi}(\cdot \mid x)$ and $z \sim \mathcal{N}(0, \sigma^2 I)$. Then, for any $\epsilon \in (0, \mathcal{R})$, if $\sigma^2 \simeq \mathcal{R}^{2\beta/d}$, there exists a comparator policy $\pi_\epsilon^*$ satisfying the following two conditions:*

$$(i) \ \mathbb{E}_{x \sim P_X}\left[r^*(x) - \mathbb{E}_{y \sim \pi_\epsilon^*(\cdot|x)}[r^\circ(x,y)]\right] \leq \epsilon, \qquad (ii) \ \mathcal{C}(\pi_\epsilon^*, \check{\pi}) \leq \epsilon^{-\gamma}\mathcal{R}^\beta.$$

*Proof.* As same as Lemma 19, we set the policy $\pi_\epsilon^*$ as

$$\pi_\epsilon^*(y \mid x) := \frac{\mathbb{1}_{S_\epsilon(x)}(y)}{\lambda(S_\epsilon(x))}.$$

The condition (i) can be confirmed by the totally same calculation as Lemma 19. To discuss the condition (ii), we first lower bound the density of $\check{\pi}$. For $y \in S_\epsilon(x)$, we have

$$\check{\pi}(y \mid x) = \frac{1}{(2\pi\sigma^2)^{d/2}} \int \widehat{\pi}(z \mid x) \exp\left(-\frac{\|y - z\|^2}{2\sigma^2}\right) dz$$

$$\geq \frac{1}{(2\pi\sigma^2)^{d/2}} \int_{S_{2C\mathcal{R}}(x)} \widehat{\pi}(z \mid x) \exp\left(-\frac{\|y - z\|^2}{2\sigma^2}\right) dz.$$

For $y \in S_\epsilon(x), z \in S_{2C\mathcal{R}}(x)$, it holds that $\|y - z\| \leq \|y\| + \|z\| \leq \epsilon^{\beta/d} + (2C\mathcal{R})^{\beta/d}$ by **(A1)**. Therefore, we have

$$\check{\pi}(y \mid x) \geq \frac{1}{(2\pi\sigma^2)^{d/2}} \exp\left(-\frac{(\epsilon^{\beta/d} + (2C\mathcal{R})^{\beta/d})^2}{2\sigma^2}\right) \int_{S_{2C\mathcal{R}}(x)} \widehat{\pi}(z \mid x) dz$$

$$\geq \frac{1}{(2\pi\sigma^2)^{d/2}} \exp\left(-\frac{(\epsilon^{\beta/d} + (2C\mathcal{R})^{\beta/d})^2}{2\sigma^2}\right) \cdot \mathbb{P}_{y \sim \widehat{\pi}(\cdot|x)}[r^*(x) - r^\circ(x,y) \leq 2C\mathcal{R}]$$

$$\geq \frac{1/2}{(2\pi\sigma^2)^{d/2}} \exp\left(-\frac{9\mathcal{R}^{2\beta/d}}{2\sigma^2}\right).$$

In the third inequality, we used the fact that

$$\mathbb{P}_{y \sim \widehat{\pi}(\cdot|x)}[r^*(x) - r^\circ(x,y) \leq 2C\mathcal{R}] = 1 - \mathbb{P}_{y \sim \widehat{\pi}(\cdot|x)}[r^*(x) - r^\circ(x,y) > 2C\mathcal{R}]$$

$$\geq 1 - \frac{C\mathcal{R}}{2C\mathcal{R}} = \frac{1}{2}.$$

By setting $\sigma^2 = 9\mathcal{R}^{2\beta/d}/2$, we have

$$\check{\pi}(y \mid x) \gtrsim \frac{1}{(2\pi\sigma^2)^{d/2}} \exp(-1) \gtrsim \mathcal{R}^{-\beta}.$$

Therefore, we have

$$\mathcal{C}(x; \pi_\epsilon^*, \check{\pi}) = \mathbb{E}_{y \sim \pi_\epsilon^*(\cdot|x)}\left[\frac{\pi_\epsilon^*(y \mid x)}{\check{\pi}(y \mid x)}\right]$$

$$= \int \frac{\pi_\epsilon^*(y \mid x)^2}{\check{\pi}(y \mid x)} dy$$

$$\lesssim \int \frac{\mathbb{1}_{S_\epsilon(x)}(y)/\lambda(S_\epsilon(x))^2}{\mathcal{R}^{-\beta}} dy$$

$$\lesssim \frac{\mathcal{R}^\beta}{\lambda(S_\epsilon(x))} \lesssim \mathcal{R}^\beta \epsilon^{-\gamma},$$

which completes the proof. □

*Proof of Theorem 7.* We define $\epsilon_{\mathtt{RM}}^{(\tau)}(x) := \mathbb{E}_{y \sim \pi^{(\tau-1)}(\cdot|x)}[(\widehat{r}^{(\tau)}(x,y) - r^{\circ}(x,y))^2]^{1/2}$ and $\epsilon_{\mathtt{RM}}^{(\tau)} := \left( \mathbb{E}_{x \sim P_X}[\epsilon_{\mathtt{RM}}^{(\tau)}(x)^2] \right)^{1/2}$. Moreover, let $\pi_{\bullet}^{(\tau)}$ be the policy which is the pure output of `InferenceTimePessimism`, i.e., the distribution of samples drawn from `InferenceTimePessimism` before adding Gaussian noises. We note that $1 + E_\tau \cdot \frac{2}{2+\gamma} \cdot \beta u_{\widetilde{s},p} = E_{\tau+1}$.

We first upper-bound the probability that there exists a step index $\tau \in [T]$ such that $|\mathcal{T}_\tau| < \frac{n_0}{2 \cdot 5^d}$. Suppose that $\sigma^{(\tau)} \le 1$ for all $\tau$. Later, we indeed choose $\sigma^{(\tau)}$ to satisfy this. For $a \in [0,1]^d$ and $X \sim \mathcal{N}(a, I_d)$, we have

$$\mathbb{P}[X \in [0,1]^d] = (\mathbb{P}[X_1 \in [0,1]])^d \ge \left( \frac{1}{\sqrt{2\pi}} \exp\left( -\frac{1^2}{2} \right) \right)^d \ge \frac{1}{5^d}.$$

Then, we have $\mathbb{P}_{x \sim \pi^{(\tau)}}[x \in [0,1]^d] \ge \frac{1}{5^d}$. Therefore, for all $\tau \in [T]$, we have

$$\mathbb{P}\left[ |\mathcal{T}_\tau| \le \frac{n_0}{2 \cdot 5^d} \right] \lesssim \exp\left( -\frac{n_0 \cdot 1/5^d \cdot (1-1/2)^2}{2} \right) = \exp\left( -\frac{n}{8 \cdot 5^d \cdot \log n} \right).$$

Hence, we have

$$\mathbb{P}\left[ \forall \tau \in [T], |\mathcal{T}_\tau| \le \frac{n_0}{2 \cdot 5^d} \right] \lesssim \exp\left( -\frac{n}{8 \cdot 5^d \cdot \log n} \right) \cdot \log n \lesssim \mathrm{e}^{-\sqrt{n}}.$$

Therefore, the regret can be bounded as

$$\mathbb{E}\left[ \mathbb{E}_{x \sim P_X}[r^*(x) - \mathbb{E}_{y \sim \pi^{(T)}}[r^{\circ}(x,y)]] \right]$$
$$\lesssim \mathbb{E}\left[ \mathbb{E}_{x \sim P_X}[r^*(x) - \mathbb{E}_{y \sim \pi^{(T)}}[r^{\circ}(x,y)]] \,\Big|\, \forall \tau \in [T], |\mathcal{T}_\tau| \ge \frac{n_0}{2 \cdot 5^d} \right] + R\mathrm{e}^{-\sqrt{n}}.$$

The following discussion is under the event that $|\mathcal{T}_\tau| \ge \frac{n_0}{2 \cdot 5^d} \gtrsim n_0$ for all $\tau \in [T]$.

For $\tau = 1$, Theorem 2 in Suzuki (2018) and Theorem 20 implies that

$$\epsilon_{\mathtt{RM}}^{(1)} = \mathbb{E}_{x \sim P_X}[\epsilon_{\mathtt{RM}}^{(1)}(x)^2]$$
$$\lesssim n_0^{-\frac{2\widetilde{s}}{2\widetilde{s}+1}} \log^2 n_0$$
$$\mathcal{R}_{\bullet}^{(1)} := \mathbb{E}_{x \sim P_X}[r^*(x) - \mathbb{E}_{y \sim \pi_{\bullet}^{(1)}(\cdot|x)}[r^{\circ}(x,y)]]$$
$$\lesssim n_0^{-\frac{2\widetilde{s}}{2\widetilde{s}+1} \cdot \frac{1}{2+\gamma}} \log^{\frac{4}{2+\gamma}} n_0.$$

Next, we derive the relation between $\mathcal{R}_{\bullet}^{(\tau)}, \mathcal{R}^{(\tau)}, \epsilon_{\mathtt{RM}}^{(\tau+1)}$ and $\mathcal{R}_{\bullet}^{(\tau+1)}$. First, we evaluate the regret $\mathcal{R}^{(\tau)}$ of the policy $\pi^{(\tau)}$. We have

$$\mathbb{E}_{y \sim \pi^{(\tau)}(\cdot|x)}[r^{\circ}(x,y)] = \frac{1}{(2\pi(\sigma^{(\tau)})^2)^{d/2}} \int \pi_{\bullet}^{(\tau)}(z \mid x) \exp\left( -\frac{\|y-z\|^2}{2(\sigma^{(\tau)})^2} \right) r^{\circ}(x,y) \, \mathrm{d}y \, \mathrm{d}z$$

$$\ge \frac{1}{(2\pi(\sigma^{(\tau)})^2)^{d/2}} \int \pi_{\bullet}^{(\tau)}(z \mid x) \exp\left( -\frac{\|y-z\|^2}{2(\sigma^{(\tau)})^2} \right) r^{\circ}(x,z) \, \mathrm{d}y \, \mathrm{d}z$$

$$- \frac{1}{(2\pi(\sigma^{(\tau)})^2)^{d/2}} \int \pi_{\bullet}^{(\tau)}(z \mid x) \exp\left( -\frac{\|y-z\|^2}{2(\sigma^{(\tau)})^2} \right) |r^{\circ}(x,y) - r^{\circ}(x,z)| \, \mathrm{d}y \, \mathrm{d}z$$

$$\ge \int \pi_{\bullet}^{(\tau)}(z \mid x) r^{\circ}(x,z) \, \mathrm{d}y \, \mathrm{d}z - \frac{1}{(2\pi(\sigma^{(\tau)})^2)^{d/2}} \int \pi_{\bullet}^{(\tau)}(z \mid x) \exp\left( -\frac{\|y\|^2}{2(\sigma^{(\tau)})^2} \right) \|y\|^\alpha \, \mathrm{d}y \, \mathrm{d}z$$

$$= \mathbb{E}_{y \sim \pi_{\bullet}^{(\tau)}(\cdot|x)}[r^{\circ}(x,y)] - C'' \cdot (\sigma^{(\tau)})^\alpha,$$

for some constant $C'' > 0$, where $\alpha := \min(\widetilde{s} - 1/p, 1)$. Therefore, we have

$$\mathcal{R}^{(\tau)} = \mathbb{E}_{x \sim P_X}[r^*(x) - \mathbb{E}_{y \sim \pi^{(\tau)}(\cdot|x)}[r^{\circ}(x,y)]] \lesssim \mathcal{R}_{\bullet}^{(\tau)} + (\sigma^{(\tau)})^\alpha.$$

Using Theorem 11, we have

$$\epsilon_{\mathrm{RM}}^{(\tau+1)} = \mathbb{E}_{x \sim P_X} \mathbb{E}_{y \sim \pi^{(\tau+1)}(\cdot|x)}[(r^{(\tau+1)}(x,y) - r^\circ(x,y))^2]$$

$$\lesssim \left[ \mathcal{R}_\bullet^{(\tau)} + (\sigma^{(\tau)})^\alpha \right]^{\frac{2\beta\varsigma}{2s+d}} n_0^{-\frac{2\tilde{s}}{2\tilde{s}+1}} \log^2(n_0).$$

By setting $\sigma^{(\tau)} \simeq (\mathcal{R}_\bullet^{(\tau)})^{2\beta/d}$, we have

$$\mathcal{R}^{(\tau)} \lesssim \left( \mathcal{R}_\bullet^{(\tau)} \right)^{\frac{2\alpha\beta}{d}}$$

$$\epsilon_{\mathrm{RM}}^{(\tau+1)} \lesssim (\mathcal{R}_\bullet^{(\tau)})^{\frac{2\beta}{d} \frac{2\alpha\beta\varsigma}{2s+d}} n_0^{-\frac{2\tilde{s}}{2\tilde{s}+1}} \log^2(n_0).$$

and Lemma 21 implies that there exists a comparator policy $\pi_{\epsilon,\tau}^*$ satisfying

$$\mathbb{E}_{x \sim P_X} \left[ r^*(x) - \mathbb{E}_{y \sim \pi_{\epsilon,\tau}^*(\cdot|x)}[r^\circ(x,y)] \right] \le \epsilon, \qquad \mathcal{C}(\pi_{\epsilon,\tau}^*, \pi_\bullet^{(\tau)}) \le \epsilon^{-\gamma}(\mathcal{R}^{(\tau)})^\beta.$$

Therefore, the same analysis as Theorem 20 implies that

$$\mathcal{R}_\bullet^{(\tau+1)} = \mathbb{E}_{x \sim P_X} \left[ r^*(x) - \mathbb{E}_{y \sim \pi_\bullet^{(\tau+1)}(\cdot|x)}[r^\circ(x,y)] \right]$$

$$\le \mathbb{E}_{x \sim P_X} \left[ r^*(x) - \mathbb{E}_{y \sim \pi_{\epsilon,\tau}^*}[r^\circ(x,y)] \right] + \mathbb{E}_{x \sim P_X} \left[ \mathbb{E}_{y \sim \pi_{\epsilon,\tau}^*}[r^\circ(x,y)] - \mathbb{E}_{y \sim \pi_\bullet^{(\tau+1)}}[r^\circ(x,y)] \right]$$

$$\lesssim \epsilon + \epsilon^{-\frac{\gamma}{2}} \cdot (\mathcal{R}^{(\tau)})^{\frac{\beta}{2}} \cdot \epsilon_{\mathrm{RM}}^{(\tau+1)}.$$

The right-hand side is minimized when $\epsilon \simeq ((\mathcal{R}^{(\tau)})^{\frac{\beta}{2}} \cdot \epsilon_{\mathrm{RM}}^{(\tau+1)})^{\frac{1}{2+\gamma}}$. Thus, we have

$$\mathcal{R}_\bullet^{(\tau+1)} \lesssim \left( \mathcal{R}_\bullet^{(\tau)} \right)^{\frac{\alpha\beta}{2+\gamma} \frac{2\beta}{d} \left( \frac{1}{2} + \frac{2\varsigma}{2\tilde{s}+1} \right)} n_0^{-\frac{1}{2+\gamma} \frac{2\tilde{s}}{2\tilde{s}+1}} \log^{\frac{4}{2+\gamma}}(n_0).$$

Let $u := \frac{\alpha\beta}{2+\gamma} \frac{2\beta}{d} \left( \frac{1}{2} + \frac{2\varsigma}{2\tilde{s}+1} \right)$. Then, we have

$$\mathcal{R}_\bullet^{(T)} \lesssim n_0^{-\frac{1}{2+\gamma} \frac{2\tilde{s}}{2\tilde{s}+1} \cdot (1+u+\cdots+u^{T-1})} \operatorname{poly} \log(n)$$

$$\lesssim \left( \frac{\log n}{n} \right)^{\frac{1}{2+\gamma} \frac{2\tilde{s}}{2\tilde{s}+1} \frac{1-u^T}{1-u}} \operatorname{poly} \log(n).$$

For $a, b \in (0,1)$, $n^{a \cdot b^{\log n}} = e^{a \cdot n^{-\log b^{-1}} \cdot \log n}$ is convergent to 1 as $n \to \infty$, the component including $(\frac{1}{n})^{a \cdot b^{\log n}}$ is bounded by some constant. Thus, we have

$$\mathcal{R}_\bullet^{(T)} \lesssim \left( \frac{\log n}{n} \right)^{\frac{1}{2+\gamma} \frac{2\tilde{s}}{2\tilde{s}+1} \frac{1}{1-u}} \operatorname{poly} \log(n),$$

which completes the proof. $\qquad\square$

# E AUXILIARY LEMMAS

## E.1 GUARANTEES FOR THE METRIC $\rho_{\boldsymbol{p},q}$

In this section, we provide some facts on the map $\rho_{\boldsymbol{p},q}$.

**Lemma 22.** *Let $q \in (0,1]$, $p_1, \ldots, p_d \in (0, 1/q)$ be some constants. We define $\rho_{\boldsymbol{p},q} : \mathbb{R}^2 \to \mathbb{R}_{\ge 0}$ by $\rho_{\boldsymbol{p},q}(x,y) := \left( \sum_{i=1}^d |x_i - y_i|^{p_i} \right)^q$. Then, $\rho_{\boldsymbol{p},q}$ is a metric on $\mathbb{R}^d$. Moreover, the volume of the ball $B(x, r; \rho_{\boldsymbol{p},q})$ centered at $x \in \mathbb{R}^d$ with radius $r > 0$ is given by $\lambda(B(x, r; \rho_{\boldsymbol{p},q})) = C_{\boldsymbol{p},q,d} \cdot r^{\frac{1}{q} \sum_{i=1}^d \frac{1}{p_i}}$, where $C_{\boldsymbol{p},q,d}$ is a constant that only depends on $\boldsymbol{p}, q, d$.*

*Proof.* First, we show that $\rho_{\boldsymbol{p},q}$ is a metric. The symmetry and the equivalence of $\rho_{\boldsymbol{p},q} = 0$ and $x = y$ are trivial. We show the triangle inequality. Let $\rho_i := |x_i - y_i|^{p_i q}$ for $i \in [d]$. Since it holds $1/p_i q \geq 1$, for any $x, y, z \in \mathbb{R}^d$, we have

$$((x_i - y_i)^{p_i q} + (y_i - z_i)^{p_i q})^{\frac{1}{p_i q}} \geq (x_i - y_i) + (y_i - z_i) = x_i - z_i,$$

which implies $\rho_i(x_i, y_i) + \rho_i(y_i, z_i) \geq \rho_i(x_i, z_i)$. Therefore, for each $i \in [d]$, $\rho_i$ is a metric on $\mathbb{R}$. Hence, we have

$$\rho_{\boldsymbol{p},q}(x, z) = \left( \sum_{i=1}^d \rho_i(x_i, z_i)^{1/q} \right)^q \leq \left( \sum_{i=1}^d (\rho_i(x_i, y_i) + \rho_i(y_i, z_i))^{1/q} \right)^q$$

$$\leq \left( \sum_{i=1}^d \rho_i(x_i, y_i)^{1/q} \right)^q + \left( \sum_{i=1}^d \rho_i(y_i, z_i)^{1/q} \right)^q = \rho_{\boldsymbol{p},q}(x, y) + \rho_{\boldsymbol{p},q}(y, z).$$

Here, in the second inequality, we applied Minkowski's inequality with $1/q \geq 1$. This completes the proof.

Next, we consider the volume of the ball $B(x, r; \rho_{\boldsymbol{p},q})$. We have

$$\lambda(B(x, r; \rho_{\boldsymbol{p},q})) \simeq \int_{x_i > 0, \sum x_i^{p_i} \leq r^{1/q}} \mathrm{d}x_1 \cdots \mathrm{d}x_d$$

$$\simeq \int_{u_i \geq 0, \sum u_i \leq r^{1/q}} \prod_i u_i^{1/p_i - 1} \, \mathrm{d}u_1 \cdots \mathrm{d}x_d \quad (u_i := x_i^{p_i})$$

$$\simeq r^{\frac{1}{q} \sum_i \frac{1}{p_i}},$$

which completes the proof. $\qquad\qquad\square$

### E.2 APPROXIMATION POWER OF NEURAL NETWORKS

In this section, we provide an approximation of elementary functions using neural networks.

**Lemma 23** (Schmidt-Hieber (2020)). *For any $\epsilon > 0$, there exists a neural network $\phi \in \Phi(L, W, S, B)$ with*

$$L \lesssim \log(1/\epsilon), \quad W \lesssim 1, \quad S \lesssim \log(1/\epsilon), \quad B \lesssim 1,$$

*such that*

$$\sup_{x,y \in [-C,C]} |\phi(x, y) - xy| \leq \epsilon.$$

