# OpenReview forum: "Inference-time Alignment with Rewards in Anisotropic Besov Spaces: Superiority of Neural Networks over Linear Estimators"
_ICLR.cc/2026/Conference — Submitted to ICLR 2026_

### Official Review · Reviewer_YRsG · 2025-10-28

**Soundness:** 3
**Presentation:** 3
**Contribution:** 3
**Rating:** 6
**Confidence:** 3

**Summary:**

This paper makes valuable theoretical contributions to reward modeling and inference alignment, with innovative insights into neural networks’ advantages and a novel multi-step learning mechanism. However, critical gaps in experimental validation, practical applicability (strong assumptions, computational cost), and accessibility significantly limit its impact.

**Strengths:**

The paper pioneers the theoretical analysis of neural networks’ advantages in alignment during inference, and formally proves that neural networks are more suitable for reward modeling compared to linear models (e.g., linear regression, kernel methods). This provides solid theoretical support for subsequent research, such as the design of more effective inference optimization strategies.
A Multi-step Alignment mechanism is proposed, extending traditional single-step reward learning to multi-step reward learning, which enriches the methodological framework of reward modeling in related fields.

**Weaknesses:**

The paper only presents theoretical derivations without providing experimental validation to support the proposed theories. No empirical results, numerical simulations, or real-world case studies are included to verify the correctness, effectiveness, or practical relevance of the theoretical claims.
The regret reduction in the multi-step training of the final proposed reward model relies on strong conditional assumptions. These assumptions include the reward function belonging to the Besov space and independent and identically distributed (i.i.d.) samples in each round, which limit the method’s generality and practical applicability.
The multi-step update mechanism involves sampling, scoring, and training the reward model in each round. The actual computational cost is likely to grow rapidly with the number of rounds and samples, yet the paper does not discuss the potential computational overhead of the algorithm or provide complexity analysis or optimization strategies.
Neural networks are prone to overfitting under small-sample scenarios. The paper fails to address this critical issue or propose mitigation measures (e.g., regularization, data augmentation, or few-shot learning techniques) to ensure the model’s generalization performance.
The paper has a high reading threshold and does not achieve accessibility. The content lacks intuitive explanations, illustrative examples, or simplified descriptions of core concepts, which hinders the dissemination and practical application of the proposed theories, especially for researchers outside the specialized subfield.

**Questions:**

Design numerical experiments or real-world case studies to verify the theoretical claims, including the superiority of neural networks over linear models in reward modeling and the effectiveness of the Multi-step Alignment mechanism.

Either relax restrictive assumptions (e.g., extend the framework to non-Besov reward functions or non-i.i.d. samples) and provide corresponding theoretical adjustments, or clearly justify why these assumptions are necessary and discuss the method’s performance under relaxed conditions via sensitivity analysis.

---

> ### Author Response · Authors · 2025-11-23
>
> We thank the reviewer for the helpful feedback. We address the specific concerns and questions below.
>
> > The paper only presents theoretical derivations without providing experimental validation to support the proposed theories. ...
>
> > Design numerical experiments or real-world case studies to verify the theoretical claims, including the superiority of neural networks over linear models in reward modeling and the effectiveness of the Multi-step Alignment mechanism.
>
> Thank you for raising this important point. The effectiveness of inference-time alignment based on neural-network–based reward modeling has already been demonstrated in many empirical studies. The aim of our work is to provide a theoretical foundation for these existing empirical findings.
>
> > The regret reduction in the multi-step training of the final proposed reward model relies on strong conditional assumptions. These assumptions include the reward function belonging to the Besov space and independent and identically distributed (i.i.d.) samples in each round, which limit the method’s generality and practical applicability.
>
> Thank you for pointing this out. Anisotropic Besov spaces form a function class that contains many well-known spaces. In fact, as we mention right after Definition 2, they include Hölder spaces as a special case. Moreover, when ($p = q = 2$) and ($s_1 = \cdots = s_d = s$), the Besov space coincides with the Sobolev space ($W_2^s$).
>
> Regarding the i.i.d. assumption, many theoretical works on regression [Schmidt-Hieber, 2017; Suzuki, 2018; Suzuki & Nitanda, 2021] and on reward maximization [Huang et al., 2025a; Foster et al., 2025] also assume that samples are given i.i.d., so we do not view this as a particularly strong or restrictive condition in our setting.
>
> > The multi-step update mechanism involves sampling, scoring, and training the reward model in each round. The actual computational cost is likely to grow rapidly with the number of rounds and samples, yet the paper does not discuss the potential computational overhead of the algorithm or provide complexity analysis or optimization strategies.
>
> Thank you for raising this important point. As the reviewer correctly points out, the multi-step algorithm has a higher computational cost than the single-step algorithm (although the cost remains polynomial in $(\text{regret})^{-1}$). Our theoretical focus in this paper is on deriving regret upper bounds by the number of oracle queries $n$, and for this reason we did not place much emphasis on computational complexity. That said, reducing the computational overhead of the algorithm is an important problem, and we will explicitly mention this as a direction for future work.
>
> > Neural networks are prone to overfitting under small-sample scenarios. The paper fails to address this critical issue or propose mitigation measures (e.g., regularization, data augmentation, or few-shot learning techniques) to ensure the model’s generalization performance.
>
> Our analysis does not merely bound the training error; rather, we derive guarantees for learning the reward model using a finite number of samples. In particular, we assume that the neural network satisfies a certain sparsity condition (see line 228), which effectively plays the role of a regularization assumption. Under this sparsity condition, our theoretical results control the generalization error of the learned reward model, so the guarantees we obtain are not undermined by overfitting.
>
> > The paper has a high reading threshold and does not achieve accessibility.
>
> Thank you for this valuable feedback. We recognize that, based on the current exposition, it is difficult to see what kinds of functions satisfy our assumptions (in particular, Assumption 6) and how the multi-step algorithm contributes to improving the regret rate. In the revised version, we will add intuitive explanations and illustrative figures to clarify these points.
>
> > Either relax restrictive assumptions (e.g., extend the framework to non-Besov reward functions or non-i.i.d. samples) and provide corresponding theoretical adjustments, or clearly justify why these assumptions are necessary and discuss the method’s performance under relaxed conditions via sensitivity analysis.
>
> Thank you for this suggestion. As mentioned above, we believe that (anisotropic) Besov spaces already form a sufficiently rich function class, so further weakening this assumption is not of primary importance for our present goals. We also view the structural conditions in Assumptions 3 and 6 as essentially necessary for establishing our theoretical guarantees. Investigating whether these assumptions can be removed or significantly relaxed by designing different algorithms is an interesting direction for future work.
>
> We would be happy to clarify any concerns or answer any questions that may come up during the discussion period.

---

### Official Review · Reviewer_YUcg · 2025-10-30

**Soundness:** 3
**Presentation:** 2
**Contribution:** 2
**Rating:** 4
**Confidence:** 3

**Summary:**

The paper considers the problem of inference time alignment where the reward is assumed to belong to an anisotropic Besov space. The reward is modeled using a neural network, which offers better modeling capabilties than linear functions and hence can learn the reward functions better. Under this setting, the authors propose a multistage inference-time alignment algorithm based on the InferenceTimePessimissm algorithm that was recently proposed. The authors derive regret bounds for their algorithm and show that the neural network based modelling offers improved regret performance over more traditional linear function estimators. Furthermore, since the authors also show that their mutistep algorithm (as opposed to single step) further improves the regret bound.

**Strengths:**

The authors clearly establish how multi-step routine helps improve their algorithm over a single step. This seems to be an interesting result.

Moreover, the role of $\beta$ is interesting. It clearly shows how local smoothness near the maximum affects convergence.

**Weaknesses:**

(This is more of a question than a statement but I would like to mention it separately.)

I am still a bit confused about the motivation and novelty of the paper. So as mentioned, the paper focuses on answering the question "What advantages do neural networks offer for inference time alignment and how can we unlock their full potential?"

To answer this, the authors show that when the reward function is in an anisotropic Besov space, NNs are better than linear estimators at modeling it and hence perform better. This has nothing to do with the inference time alignment --- this is a question about reward modeling. Moreover, the superiority of NNs over linear estimators was already established for learning functions in the paper  (Suzuki & Nitanda, 2021). The authors perform a $L_2 - L_{\infty}$ bound transformation in this work but that is relatively quite minor. So the benefit of NNs is already established.

In terms of application to inference-time alignment, the performance clearly depends on reward estimation error. The authors build their algorithm on InferenceTimePessimissm, which clearly outlines the role of estimation error. So, one can easily plug and play the improved result. With these results, it almost seems that the main answer they are looking for is a plug and play from these results.

Of course there is a multi-stage algorithm that is novel, but it is not central to the question. And I agree that part is definitely novel and does not follow plug and play.

Am I missing something very basic in the premise and the main results or this is pretty much it?

**Questions:**

I have several questions/comments about the paper.

- I think the paper needs to work quite a bit on notation. There are several overloaded variables like $p$ (distribution and Besov space parameter), $r$ (radius and Besov parameter), $\tau$ (error bound, pointer in algorithm), $\mathcal{N}$ (covering number and Gaussian). This makes the paper much more difficult to read. Even in the appendix, the notations defined in the paper are not used as is, e.g., for a ball (the metric is often skipped). I think this requires a detailed revision of the paper.

- I am curious about the assumption that the reward function belongs to an anistropic Besov space. These spaces are extremely general function spaces that encode minimum regularity requirements in a sense. They are beneficial to characterize the class where NNs are better than linear function (because one is searching for general classes). Real world functions tend to offer more regularity than this general class and while the generality subsumes such functions, it fails to leverage their additional structure. In fact, the benefit over linear functions is only realized in a regime with $p \in [1, 2)$. Given that the advantage is only to be seen in specific subset of functions, that appear more often in mathematical constructs than they do in real life, I am curious about scenarios in real-world applications, where such a generality *actually* helps.

- In addition to the above question, most of the results omit leading constants and hold for large constants and sample sizes. Are such sample sizes relevant in inference time alignment, where typically we don't have very large datasets?

- Moreover, in typical application scenarios both $\Omega_X$ and $\Omega_Y$ are finite. How does one interpret these results for typical application scenarios like these since the results heavily rely on the continuity of the domain?

- In the definition of anisotropic Besov space, shouldn't $r$ be $\min \lfloor s_i \rfloor$? Typically, in fractionally smooth spaces, the derivative that is required to integrable is the greatest integer smaller than the smoothness parameter of the space.

- In Assumption 6, what do you mean by $s > d/p$. $s$ is a vector so that inequality does not make sense. Did you mean $\tilde{s}$?

- In Assumption 6, A1 is it not too strong to assume that the result holds for all $\epsilon > 0$. Firstly, $\epsilon \in (0,1)$ is a necessary requirement for the inclusion to hold. Secondly, the way the assumption is stated, it seems to be global property. Usually it is more reasonable to assume that it holds locally in a neighborhood.

- In Assumption 6, A2, what is $r$?

- In Lemma 19, you state there is an A3 of Assumption 6? I couldn't find any such assumption.

- In the mollification step, what do you do when $y$ goes beyond the domain $\Omega_Y$? Since $\Omega_Y$ is bounded but Gaussian is not, so it is important to handle such cases. Moreover, does that affect the analysis?

- What is the choice of $\sigma^{(\tau)}$? I see a value of $\tau^{\beta/d}$ in the appendix, but I could not find it in the main paper? What is the motivation behind this choice (except for the math working out)?

---

> ### Author Response · Authors · 2025-11-23
>
> We thank the reviewer for the helpful feedback. We address the specific concerns and questions below.
>
> > I am still a bit confused about the motivation and novelty of the paper. ... This has nothing to do with the inference time alignment --- this is a question about reward modeling.
>
> We agree that part of our analysis can be viewed as a question about reward modeling. However, we consider reward modeling is a central problem in inference-time alignment because improving the accuracy of the reward model directly improves the performance of inference-time alignment. In our work, we do not only show that neural networks approximate rewards better than linear estimators, but also show that this improved approximation translates into strictly better *regret* guarantees. In this sense, our results go beyond pure reward modeling and provide end-to-end guarantees for alignment performance.
>
> Moreover, Foster et al. (2025) also emphasize that learning a good reward model is a central challenge for inference-time compute, i.e., for maximizing a reward using a pretrained model at inference time. Our analysis is aligned with this perspective.
>
> > Moreover, the superiority of NNs over linear estimators was already established for learning functions in the paper (Suzuki & Nitanda, 2021). ... Of course there is a multi-stage algorithm that is novel, but it is not central to the question.
>
> As the reviewer correctly points out, our main contribution regarding the upper bounds achieved by neural networks lies in the proposal and analysis of the *multi-step algorithm*. The regret upper bound for neural networks in the single-step algorithm is essentially obtained by combining the results of Huang et al. (2025a) and Suzuki & Nitanda (2021) (, although there is a step where we adapt the result of Huang et al. (2025a) to our setting). In a multi-step algorithm, it is necessary to increase the expected reward at each step while ensuring that the coverage does not become too large. Therefore, our algorithm and its analysis exhibit substantially greater novelty than a mere repeated application of a single-step analysis. This contribution presents a way to maximize the capabilities of neural networks in inference-time alignment, which provides an answer to the question posed in this paper.
>
> On the other hand, we regard the single-step upper bound as an important preliminary stage for the multi-step algorithm.
> The multi-step algorithm in Section 4 alternates between a modified version of the algorithm presented in Huang et al. (2025a) and learning the reward model.
> We believe that including the single-step result makes the paper easier to follow.
> It is also necessary in order to present a comparison with the limitations of linear estimators. For these reasons, we chose to first explain the single-step neural-network upper bound in Section 3.1.
>
> > I think the paper needs to work quite a bit on notation. ...
>
> Thank you for pointing out the overloading of notation.
> We agree that this hurts readability and are working on the revision throughout the paper and appendix.
> In particular, we will
> (i) replace the use of $p$ for the distribution by $\pi$ (e.g., around line 134 and line 251), reserving $p$ only for the Besov parameter.
> (ii) use distinct symbols for the radius and the Besov parameter instead of overloading $r$;
> (iii) use $\tau$ only as the index of the step in the algorithm and rename the error bound to a different symbol.
> (iv) use a different letter $\mathcal{M}$ for covering numbers to clearly distinguish it from the Gaussian ($\mathcal{N}$).
>
> We will also make the definition of balls consistent in the appendix.
> Specifically, we will explicitly include the notation for the metric that was omitted in the definitions of the ball, its volume, and the covering number.

---

> ### Author Response · Authors · 2025-11-23
> **(Continued from the previous comment)**
>
> > I am curious about the assumption that the reward function belongs to an anistropic Besov space. ...
>
> To the best of our knowledge, this work is the first to analyze reward maximization problems solved by neural networks in the inference-time alignment setting. For such an initial study, we believe it is natural to work in a sufficiently general functional framework.
>
> Extending our results to more specific structural assumptions that may better capture particular application domains is, in our view, an important and meaningful direction for future work. However, if one starts from the outset with very special structure, there is a risk that the intrinsic difference in capability between neural networks and linear estimators becomes obscured by problem-specific details. We see it as a key contribution of our work that, even without imposing such specialized structure, we can already exhibit a gap between neural and linear estimators in this general setting.
>
> Furthermore, we believe that the regime $p\in[1,2)$ is precisely the one that is highly relevant for real-world problems. When $p\in[1,2)$, the smoothness of functions in the class is guaranteed only in an average sense over the domain, hence the function can be bumpy around some points (we explicitly discuss this right after Definition 2). It is natural to expect that real-world reward functions may contain such bumps, and our results show that in such realistic settings, neural networks can have an advantage over linear estimators.
>
> > In addition to the above question, most of the results omit leading constants and hold for large constants and sample sizes. Are such sample sizes relevant in inference time alignment, where typically we don't have very large datasets?
>
> We agree that the number of data available for inference-time alignment is typically smaller than in pre-training. However, since we are explicitly studying the learning of neural-network reward models, it is natural to assume a regime where a reasonably large number of samples is available. The theory on regression using neural networks has been studied since well before the recent rise in popularity of foundation models, and it has been standard in that literature to focus on the exponent of the sample size $n$ and to de-emphasize constant factors. Therefore, we believe that such analyses are still meaningful even when the dataset size is smaller than in pre-training.
>
> > Moreover, in typical application scenarios both $\Omega_X$ and $\Omega_Y$ are finite. How does one interpret these results for typical application scenarios like these since the results heavily rely on the continuity of the domain?
>
> We agree that in many practical applications, such as language tasks, the input and output spaces are finite. On the other hand, in real-world reasoning tasks, both the questions and the answers have infinitely many possibilities and essentially exhibit a continuous structure. It is therefore quite natural and useful from an application perspective to work in a continuous representation space that also encompasses this continuous regime. Moreover, the setting of our analysis can be viewed as the case where these discrete objects are embedded into a continuous space by an encoder, and we study the reward as a function defined on this continuous representation space. Directly treating the input and output as token sequences and modeling the reward with architectures such as Transformers is an important direction for future work.
>
> > In the definition of anisotropic Besov space, shouldn't $r$ be $\min\lceil c_i\rceil$?
>
> Thank you for pointing out this typo. It is correct that it should be $\min_i \lceil c_i \rceil$. We will correct this in the revised version.
>
> > In Assumption 6, what do you mean by $s>d/p$.
>
> Thank you also for this remark. The correct condition is $\tilde{s} > 1/p$. We will fix this in the revised version.
>
> > In Assumption 6, A1 is it not too strong to assume that the result holds for all $\epsilon>0$.
>
> The conditions in Assumption 3 and Assumption 6 can indeed be relaxed so that they are required only for $\epsilon \in (0, \epsilon_0)$ for some $\epsilon_0 > 0$, rather than for all $\epsilon > 0$. We are currently revising the paper to make this modification explicit.
>
> > In Assumption 6, A2, what is $r$?
>
> In Assumption 6 (A2), the correct expression is $V_d(\delta)$, not $V_d(r)$. This assumption states that the $\delta$-covering number of the set $S_\epsilon$ is of the same order as the ratio between the volume of $S_\epsilon$ and the volume of a ball of radius $\delta$. We will correct this in the revised version.
>
> > In Lemma 19, you state there is an A3 of Assumption 6? I couldn't find any such assumption.
>
> Thank you for catching this. In Lemma 19, the reference should be to (A1) of Assumption 6, not (A3). We will correct this in the revised version.

---

> ### Author Response · Authors · 2025-11-23
> **(Continued from the previous comment)**
>
> > In the mollification step, what do you do when goes beyond the domain $\Omega_Y$? Since $\Omega_Y$ is bounded but Gaussian is not, so it is important to handle such cases. Moreover, does that affect the analysis?
>
> Thank you for this important observation. We indeed need to discard samples that fall outside $\Omega_Y$. More concretely, between lines 5 and 6 of Algorithm 2, we should exclude those indices $t$ for which $y_t \notin \Omega_Y$. Since a sample $y$ drawn from $\pi^{(\tau)}$ belongs to $\Omega_Y$ with a constant probability, a constant proportion of the $n_0$ samples will, with high probability, lie inside $\Omega_Y$. Therefore, the regret rate is unchanged up to constant factors. We are currently revising the paper to clarify this point.
>
> > What is the choice of $\sigma^{(\tau)}$? I see a value of in the appendix, but I could not find it in the main paper? What is the motivation behind this choice (except for the math working out)?
>
> Thank you for pointing out this important issue. In the revised version, we will also describe the choice of $\sigma^{(\tau)}$ in the main text, together with a brief intuitive explanation.
>
> We choose the Gaussian noise scale $\sigma^{(\tau)}$ so as to balance two competing goals: exploiting the current reward model while maintaining sufficient coverage. Intuitively, the Gaussian noise is added to ensure that we can still explore regions that may have high true reward, while concentrating sampling around points that look promising under the current reward model. If $\sigma^{(\tau)}$ is too small, the sampling distribution can become overly concentrated around the (possibly biased) maximizers of the previous-step reward model, and we may fail to visit regions where the true reward is actually high due to estimation error. On the other hand, if $\sigma^{(\tau)}$ is too large, the samples become almost independent of the learned reward model, and we cannot effectively leverage the information gained so far. The specific choice of $\sigma^{(\tau)}$ given in the appendix is designed to strike this balance.
>
> We would be happy to clarify any concerns or answer any questions that may come up during the discussion period. We would greatly appreciate it if you could consider increasing the score once all concerns have been resolved.

---

### Official Review · Reviewer_gstx · 2025-11-01

**Soundness:** 3
**Presentation:** 3
**Contribution:** 3
**Rating:** 6
**Confidence:** 3

**Summary:**

The paper studies inference-time alignment when the reward is learned and the true reward lies in anisotropic Besov spaces. It proves (i) a regret upper bound for inference-time alignment when the reward is estimated by a sparse ReLU network (Theorem 4), (ii) a lower bound showing linear estimators are rate-suboptimal in this setting (Theorem 5), and (iii) an improvement from a multi-step procedure with Gaussian perturbation that maintains coverage and yields a sharper regret rate (Theorem 7).

**Strengths:**

Clear theoretical story linking feature learning to regret. The paper formalizes why neural networks help in inference-time alignment: better L2 estimation over anisotropic Besov classes translates (via volume/coverage conditions) into better regret than linear estimators. Theorem 5 crisply separates the classes.

Concrete algorithmic instantiation. The work grounds the analysis in the chi square-regularized Inference Time Pessimism sampler (Algorithm 1) and uses a standard coverage–estimation trade-off inequality as the backbone (Eq. (2)), which is appropriate for inference-time alignment.

**Weaknesses:**

Strength of structural assumptions. The main gains hinge on Assumption 3 (non-negligible super-level set volume) and, in the multi-step analysis, Assumptions (A1)–(A2) in Assumption 6 that constrain the geometry/covering of $S_{\epsilon}$. These are plausible but may exclude adversarial or heavily spiked rewards in practice.

**Questions:**

Are the assumptions A1-A2  in Assumption 6 central to the works or can they be relaxed?

---

> ### Author Response · Authors · 2025-11-23
>
> We thank the reviewer for the helpful feedback. We address the specific concerns and questions below.
>
> > Strength of structural assumptions. The main gains hinge on Assumption 3 (non-negligible super-level set volume) and, in the multi-step analysis, Assumptions (A1)–(A2) in Assumption 6 that constrain the geometry/covering of $S_\epsilon$. These are plausible but may exclude adversarial or heavily spiked rewards in practice.
>
> Thank you for this important comment regarding our assumptions.
> The conditions in Assumption 3 and Assumption 6 can actually be relaxed so that they are required only for $\epsilon \in (0, \epsilon_0)$ for some threshold $\epsilon_0 > 0$, rather than for all $\epsilon > 0$. We are currently revising the paper to make this point explicit. With this modification, our theoretical results still apply even if there are spikes away from the maximizer, as long as their values are smaller than the global maximum by a fixed constant margin.
>
> We also remark that the anisotropic Besov space considered in this paper already contains functions that exhibit spike-like behavior (see lines 168–169).
>
> > Are the assumptions A1-A2 in Assumption 6 central to the works or can they be relaxed?
>
> We view these assumptions as essential for theoretically justifying our multi-step algorithm. In particular, (A2) is a standard condition in the literature on multi-step black-box optimization (e.g., Wang et al., 2018), and its necessity for obtaining nontrivial convergence guarantees has already been clarified in these prior works.
>
> Regarding (A1), our assumption is somewhat stronger than those typically used before. For example, Wang et al. (2018) impose a condition on the volume of the super-level sets, similar to our Assumption 3, whereas we additionally assume a specific decay of the reward as a function of the distance to the optimal solution. This difference stems from the fact that prior work builds algorithms around reward estimators with small $L^\infty$-risk, while our analysis is based on estimators that control the $L^2$-risk. It may be possible to replace (A1) with an alternative technical condition, but we believe that some similar structural assumptions are fundamentally necessary.
> Although Assumption (A1) is stronger than those used in prior work, it can be viewed as a relaxed condition on local convexity and smoothness around the optimum. Hence, we believe it is still sufficiently mild for practical purposes.
>
> We would be happy to clarify any concerns or answer any questions that may come up during the discussion period.

---

### Author Response · Authors · 2025-12-03
**Summary of Revisions and Responses**

Dear AC and reviewers,

We sincerely appreciate your dedicated efforts in addressing the recent incident regarding reviewer anonymity.
For the AC's reference to save your time and help you quickly understand the rebuttal process, we would like to summarize the main revisions we have made in response to the reviews.
We thank all reviewers for their insightful and constructive comments, which have been very helpful in improving the paper.

- [**Relaxation of assumptions**]  Several reviewers expressed concern that, in Assumption 3 and in conditions (A1) and (A2) of Assumption 6, requiring the assumptions to hold for all $\epsilon > 0$ might be too strong. We agree that this can be relaxed, and we now assume instead that these conditions hold for $\epsilon \in (0, \epsilon_0]$ for some threshold $\epsilon_0 > 0$. The paper has been revised to reflect this weaker and more natural formulation.
- [**Modification in the algorithm**] For Algorithm 2, there was a question about how to handle the case where the sampled responses $\lbrace y_t\rbrace_t$ in line 5 fall outside $\Omega_Y$. As clarified in our response, this issue can be resolved by adding a single line to the algorithm (line 7 in the revised version, which discards the samples with $y_t \notin \Omega_Y$). Both the algorithm and the corresponding proofs have been updated accordingly.
- [**Corrections of notation and typos**] We received comments about duplicated notation (e.g., $p, r, \tau, \mathcal N$) and typos (e.g., the definition of anisotropic Besov space, the condition in Assumption 6, the notations of balls and metrics). All such issues have been corrected in the revised version.
- [**Improved presentation and clarity**] Some reviewers found the theory and the multi-step algorithm hard to follow. To improve clarity, we have added figures (Figure 1 in the revised version) that provide intuitive explanations of the assumptions and of the multi-step procedure. We have also added an explanation in the main text about the choice of the hyperparameters $\sigma^{(\tau)}$ and the motivation behind this choice.

Furthermore, we summarize below our responses to the main questions and concerns raised by the reviewers.

- There were questions regarding the novelty of the single-step analysis in Theorem 4. We would like to emphasize that, in this work, the main contribution of our upper-bound analysis for neural networks lies in the proposal and analysis of the multi-step algorithm. The single-step upper bound is presented primarily as a building block for the multi-step procedure, to improve the clarity of exposition, and to provide a comparison against the limitations of linear estimators.
- One reviewer raised the concern that the Besov space assumption without structural assumptions might be too general. This paper is the first to analyze the use of neural networks for solving reward maximization problems in the inference-time alignment setting, and we believe that it is natural for such an initial theoretical study to be conducted in a general function space. Extending our results to more specific structural assumptions is indeed an important and interesting direction for future work to derive improved regret bounds. However, starting from highly specialized structures could obscure the fundamental differences between neural networks and linear estimators. We showed a clear performance separation even in a general setting, which we believe is an important contribution.
- A reviewer also asked about our choice to consider the case where the input space $\Omega_X$ and output space $\Omega_Y$ are continuous. While many practical applications (such as language tasks) involve finite input and output sets, in real-world inference tasks, both the prompts and the responses have infinitely many possibilities and essentially exhibit a continuous structure. Thus, our analysis in a continuous space also covers such scenarios. Moreover, our setting can be interpreted as modeling discrete objects that have been embedded into a continuous representation space via an encoder.

We are grateful to the AC and the reviewers for their time and effort in evaluating our work.

---

### Meta-Review · Area_Chair_oz2H · 2026-01-11

**Summary:**

This paper studies the theoretical foundations of inference-time alignment using reinforcement learning when neural networks are employed as reward models. Under smoothness assumptions characterized by anisotropic Besov spaces, the authors derive regret bounds for neural network–based reward estimation and show that such models can outperform linear estimators by adapting to local function structure. The analysis further claims that actively and iteratively learning the reward model during inference enables improved regret performance, providing theoretical justification for the advantages of neural networks in practical inference-time alignment settings.

**Reviewer Concerns:**

- Concern 1: Unclear motivation and novelty - reward modeling vs. inference-time alignment
Authors clarified that reward modeling is central to inference-time alignment since improved reward approximation directly translates to better regret guarantees. Main novelty lies in the multi-step algorithm that maximizes neural network capabilities while maintaining coverage. Single-step results serve as a preliminary foundation. Partially addressed - authors acknowledge single-step contribution is largely combining existing results (Huang et al. 2025a + Suzuki & Nitanda 2021), though multi-step algorithm shows genuine novelty.

- Concern 2: Overloaded notation severely hurts readability

- Concern 3: Anisotropic Besov space assumption too general/abstract for practical relevance
Authors argue generality is appropriate for initial theoretical study and that regime s < d/2 (where NN advantage emerges) is precisely the realistic case where functions exhibit "bumps." It's weakly addressed and some theoretical justification is provided, but no evidence that this captures real-world reward functions; advantage only in specific regimes < d/2.

Concern 3: Lack of experimental validation
The authors state empirical effectiveness already demonstrated in existing literature; their goal is to provide a theoretical foundation. Major gap remains - no experiments, simulations, or case studies to validate theoretical claims or demonstrate practical applicability of multi-step algorithm. Also, computational cost grows polynomially, but defer optimization to future work. Its partially addressed - assumptions defended as standard but computational complexity and scalability concerns remain unaddressed.

Concern 5: Technical errors and unclear exposition
Multiple typos/errors identified: α_k should be ⌊α_k⌋ in the Besov definition, vector inequality error in Assumption 6.A1, missing A3 in Lemma 19, undefined ε^δ in A2, mollification boundary handling unclear. It appears that the paper is not ready for publication and requires some polishing.

The paper has merit as a theoretical exercise but falls short of the impact, rigor, and completeness bar for the venue. Authors are encouraged to resubmit after addressing the concerns raised and writing is improved.

**Reviewer Scores:**

YUcg did not respond and kept the score to 4; based on the limitations of the rebuttal, I believe there is not enough enthusiasm to champion for the better.

---

### Decision · Program_Chairs · 2026-01-26

Reject